# Structure Development in List-Sorting Transformers

## Abstract

We study how a one-layer attention-only transformer develops relevant structures while learning to sort lists of numbers. At the end of training, the model organizes its attention heads in two main modes that we refer to as vocabulary-splitting and copy-suppression. Both represent simpler modes than having multiple heads handle overlapping ranges of numbers. Interestingly, vocabulary-splitting is present regardless of whether we use weight decay, a common regularization technique thought to drive simplification, supporting the thesis that neural networks naturally prefer simpler solutions. We relate copy-suppression to a mechanism in `GPT-2` and investigate its functional role in our model. Guided by insights from a developmental analysis of the model, we identify features in the training data that drive the model's acquired final solution. This provides a concrete example of how the training data shape the internal organization of transformers, paving the way for future studies that could help us better understand how LLMs develop their internal structures.

## 1 Introduction

The rapid advancement of capabilities in state-of-the-art deep learning models has significantly outpaced our understanding of their internal mechanisms. This disparity poses a critical challenge for the AI community, as the deployment of increasingly powerful yet opaque models raises concerns about reliability, safety, and ethical implications. This work presents a theoretical analysis of a simplified transformer model and aims to provide insights for interpreting toy-models and understanding what drives learned solutions.

A functional understanding of complex deep learning models is a difficult task when we don't know the fundamental building blocks that the model implements. **Mechanistic interpretability** addresses this challenge by aiming for a mechanistic understanding of the model, usually by reverse-engineering it and decomposing the functional structures into circuits. Most research in this field focuses on small language models (Wang et al., 2022; Hanna et al., 2023; Gould et al., 2023), with a recent breakthrough by Templeton et al. (2024) finding interpretable features in `Claude-3` using Sparse Auto Encoders (SAEs) (Cunningham et al., 2023; Bricken et al., 2023).

Motivated by research in this field, **we study a simple yet fundamental question: How do neural networks develop organized internal structures as they learn?** We answer this question for a single-layer attention-only transformer model trained on the task of sorting lists of numbers. Its simplicity provides a controlled environment to study the impact of various hyperparameters on the learning dynamics, in ways that would be impossible with larger models.

The model was originally proposed by McDougall (2023a) and interpreted by McDougall (2023b), by mechanistically decomposing the final solution found during training in **Output-Value (OV)** and **Query-Key (QK) circuits** Elhage et al. (2021). These circuits are constructed by suitably multiplying the respective matrices in transformers with embedding and unembedding matrices. McDougall (2023b) found that when sorting lists of numbers, these circuits tend to different roles: The **QK circuit guides the model's attention** by having each input number token attend primarily to the nearest number token in the vocabulary that is slightly larger than itself, resulting in higher values along the diagonal of the QK matrix. The **OV circuit acts as a copying circuit**, copying forth number tokens that are present in the context, which results in higher values on the diagonal of the OV matrix, as opposed to the off-diagonal values. Put together, the QK and OV circuits bring attention to the smallest number token in the context, larger than

the current number token that was given as an input. Since the context consists of the unsorted list and its corresponding sorted list up to and including the current number token, the attended to number token will be the smallest number token in the unsorted list larger than the current number token. We have illustrated this sorting process in Fig. 1.

A natural question arises: Why study such a simplified model when state-of-the-art systems are orders of magnitude more complex? A compelling hypothesis that offers a potential way forward is what's called the **universality hypothesis** (Olah et al., 2020) or "convergent learning" hypothesis (Li et al., 2016). This hypothesis suggests that certain fundamental patterns in how neural networks organize information occur across different scales and architectures. Just as biologists study simple model organisms to understand principles that apply to more complex life forms, studying simple neural networks may reveal insights that generalize to larger models.

We find support for this hypothesis in our work. Our analysis of the model's final solutions revealed the presence of attention heads exhibiting characteristics analogous to the **copy-suppressing** attention heads first identified in `GPT-2` (McDougall et al., 2023). These attention heads, instead of propagating number tokens forward (as evidenced by a positive diagonal in the OV matrix), actively suppress the copying mechanism, resulting in a negative diagonal in the OV matrix. The original `GPT-2` study found evidence suggesting that copy-suppression contributes to the "self-repair" phenomenon (McGrath et al., 2023), where neural network components compensate for ablations or perturbations in other parts of the network. The role it plays in `GPT-2` as well as its significance as a structure that occurs across model sizes, makes copy-suppression an interesting and important structure for interpretability research. We investigate how it arises in our model and what functional role it fulfills.

The universality hypothesis also provides a framework for investigating a fundamental question: What factors drive the development of these organized internal structures? Is it primarily the training data, the model architecture, or specific training parameters? Through careful experimentation with our controlled setup, we identify specific patterns in the training data that influence how the model organizes its attention heads. In particular, we find that the typical gap between adjacent sorted numbers and how much these gaps vary determine whether the model develops different types of specialized components.

Complementing this mechanistic understanding, we adopt a **developmental interpretability** approach, drawing inspiration from biological analogues such as embryonic development. This perspective examines how distinct developmental stages and structures emerge during training. Recent work (Chen et al., 2023; Furman & Lau, 2024; Hoogland et al., 2024; Wang et al., 2024) has begun applying tools from Singular Learning Theory (SLT, Watanabe 2009) to establish more rigorous methods for analyzing these developmental processes. The Local Learning Coefficient (LLC) (Lau et al., 2023), for instance, provides a mathematical measure of solution complexity that helps us track how the model's internal organization evolves.

In our list-sorting model, we observe that attention heads tend to develop specialized roles during training. One of these is what we call **vocabulary-splitting** (also observed by McDougall (2023b)), where the model organizes its heads, by dividing the available number vocabulary into separate ranges for the different heads to handle. Intriguingly, this division of labor between heads corresponds to a decrease in solution complexity when compared to an earlier stage during training where both heads attend to overlapping number ranges, as measured by the LLC. This decrease in complexity persists even when we remove weight decay - a common technique thought to encourage simpler solutions, which suggests that neural networks may naturally prefer simpler solutions even without explicit pressure to do so.

Our work contributes to interpretability research at the intersection of mechanistic and developmental approaches. We summarize our main contributions below:

1. **Studying distinct developmental stages during training**, in particular two forms of head specialization:

    - **Vocabulary-splitting**, where models split the numbers in the lists (i.e. the vocabulary) into non-overlapping regions between attention heads. We find that this specialization represents a simpler solution (as measured by the LLC) compared to having multiple heads handle overlapping ranges. Interestingly, this simplification occurs naturally during training and persists

even without weight decay, suggesting that neural networks may have an inherent bias toward simpler solutions.

- **Copy-suppression**, where models develop *copying* heads that copy forth tokens as part of the learned solution to sort the list, and *copy-suppressing* heads that counteract the copying circuits. This constitutes a minimal example of copy-suppression (previously identified in `GPT-2`(McDougall et al., 2023)[1], which can be investigated more thoroughly in our simpler setting. Our findings add nuance to previous hypotheses about copy-suppression: while we confirm that these heads act to calibrate copying behavior, we find that in our case they actually increase rather than decrease model confidence, suggesting that they more broadly calibrate the copying head, reducing over-confidence if the accuracy is low like in `GPT-2`, and increasing confidence if the accuracy is high like in our case. These findings contextualize our toy-model's relevance for a more complex model and provide nuance for the functional role of copy-suppression across models.

2. **We identify one of the main drivers of different head specializations**, such as vocabulary-splitting or copy-suppression: a feature in the training dataset, namely the distribution over the gaps between adjacent list elements in the sorted lists. We characterize this distribution with its mean and variance, and probe the hypothesis by extensive variations of the training datasets. This exemplifies how seemingly minor features of the training data can fundamentally shape how neural networks organize internally, providing a concrete example of how training data characteristics shape training dynamics and learned solutions. These findings pave the way for future studies that could help us better understand how LLMs develop their internal structures.

We describe our model setup and methods in Sec. 2, present our main findings in Sec 3 with additional results in Appendices B-C, discuss the implications in Sec. 4, summarize related work and limitations of our study in Sec. 5-6 and conclude in Sec. 7.

## 2 Methods

In this section, we introduce our experimental setup and the key metrics we use to study how neural networks develop organized structures while sorting lists. First, we describe our baseline model architecture and explain how we vary different aspects of the model and training data to understand what drives the development of different organizational patterns. Then, we outline the quantitative measures we use to track the model's development, including mathematical tools that help us measure solution complexity and understand how different parts of the model work together.

### 2.1 Baseline Model Setup and Training

Following McDougall (2023a) we train a single-layer attention-only transformer model on input sequences of the form [8, 3, 5 SEP, 3, 5, 8], where numbers are sampled uniformly from 0 to 50 and do not repeat, producing a vocabulary size of 52. The model sorts by outputting the next number starting at the separation token, producing a list of numbers of the form [x, x, x, 3, 5, 8, x], where the positions marked with x are not included in the loss function.

We define the model with 2-heads, a list length of 10 numbers and a vocabulary size of 52 tokens as our **baseline model**. It includes a residual stream size of 96, two attention heads with head dimension of 48, and Layer Normalization (LN, Ba et al. 2016). The model is trained with Weight Decay (WD) set to 0.005 using the Adam optimizer (Loshchilov & Hutter, 2019) with a learning rate of $10^{-3}$, a dataset size of 150000 with a batch size of 512 and a cross-entropy loss function. The architecture is implemented using `TransformerLens v.2.1.0` (Nanda & Bloom, 2022). The hardware used for training the models are NVIDIA RTX-4060 and NVIDIA RTX-4090.

---

[1]In our case the copying and copy-suppressing head is in the same layer, whereas in `GPT-2` the copy-suppressing head is in a later layer than the copying head. This difference might be important, see the discussion in sec. 4.3.)

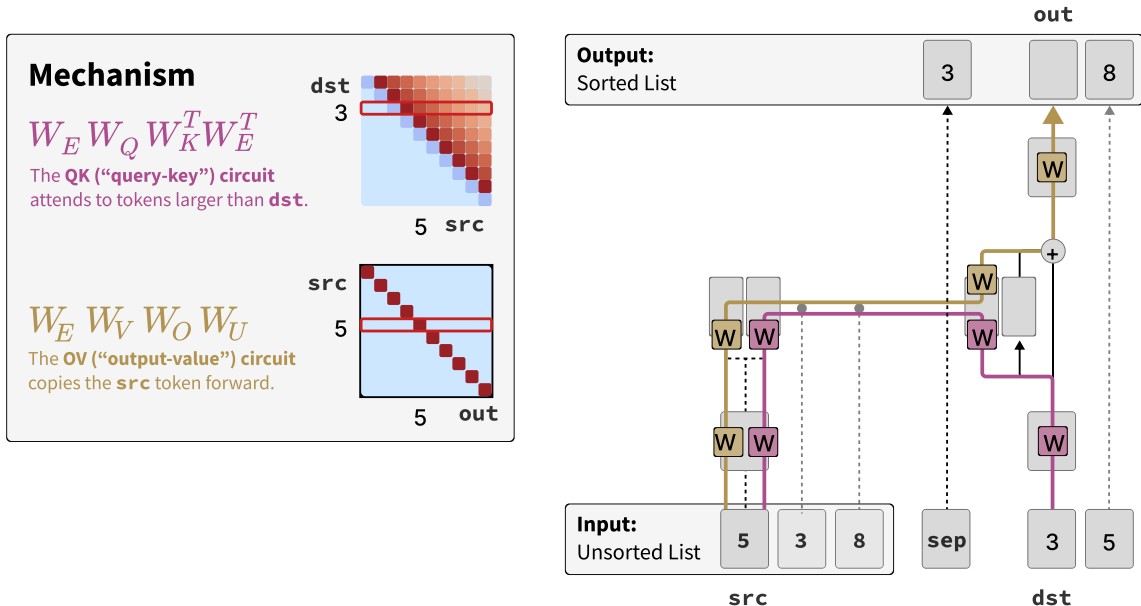

Figure 1: Illustration of the transformer architecture and an idealized version of the sorting circuits based on a similar figure in Elhage et al. (2021). The Output-Value (OV) circuit copies forward the tokens in the context, whereas the Query-Key (QK) circuit attends more to smaller numbers larger than the current token.

We investigate several aspects of this model by **varying the baseline setup** with respect to the **architecture of the model** (number of heads, presence of LN), **training hyperparameters** (presence of WD) and **features of the training data** (list length, the vocabulary size or by manipulating the training data distribution). Regarding the latter, we find that most of the impact of varying the training data can be boiled down to changes in the distribution of the separation between list elements in the training dataset, denoted by $\delta_i = l_{i+1} - l_i$, where $l_i$ denotes the ith element of the sorted list. For example, [8, 3, 5, SEP, 3, 5, 8] has the $\delta$ values [2, 3]. We find that the distribution of $\delta$-values in the training data is important for the final solution that the model implements, and we shall denote a dataset D with mean $\delta$ value of $x$ as $D_{\overline{\delta}=x}$, where $\overline{\delta}$ stands for the mean $\delta$. Specifically, $\overline{\delta}$ is calculated as

$$\overline{\delta} = \frac{\sum_{\text{lists}} \sum_{i=1}^{\ell-1} \delta_i}{N_{\text{lists}}(\ell - 1)} = \frac{\sum_{\text{lists}} \sum_{i=1}^{\ell-1} l_{i+1} - l_i}{N_{\text{lists}}(\ell - 1)}, \tag{1}$$

where $\ell$ denotes the length of the lists, and the dataset contains $N_{\text{lists}}$.

For reference, the uniform sampling described at the beginning of this section produces $D_{\overline{\delta}\approx 4.7}$. We vary the dataset, and by consequence the distribution of $\delta$, by varying features in the training data:

1. **We vary the list length.** A dataset with a list length different from 10 indicates this in the superscript. For example, $D_{\overline{\delta}\approx 2.5}^{\ell=20}$ denotes a dataset generated with a uniform sampling as described in the beginning of this section with a list length of 20, resulting in $\overline{\delta} \approx 2.5$. We omit the superscript for the default list length 10: $D_{\overline{\delta}\approx 4.7} \equiv D_{\overline{\delta}\approx 4.7}^{\ell=10}$.

2. **We vary the vocabulary size.** A dataset with a vocabulary size different from 52 indicates this in the superscript. For example, $D_{\overline{\delta}\approx 18.3}^{v=202}$ denotes a dataset generated with uniform sampling with a vocabulary size of 202, resulting in $\overline{\delta} \approx 18.3$.

3. **We manipulate the list distribution.** We do this by starting with a dataset distributed as $D_{\overline{\delta}\approx 4.7}$, and then iteratively removing the highest $\delta$ lists with probability $(\min [\delta_l / \max_{\text{dataset}} [\delta_l] - 70\%, 0]) / 30\%$ (where $\delta_l$ is the mean $\delta$ across a single list), until the $\overline{\delta}$

reaches the desired value. We indicate that a dataset has been generated in this way by adding a $d$ to the superscript, such as $D_{\bar{\delta}\approx 2.2}^{d}$.

4. **We fix allowed $\delta$ values.** In addition to manipulating the mean of the distribution $\bar{\delta}$, we also want to vary the spread. To this end, we construct the lists by uniformly sampling 9 $\delta_i$ allowed values, producing a sorted list with elements $l_i$, with $l_0 = 0$ and $l_{i+1} = \text{cumsum}(\delta_i)$. If $\max_i[l_i] > 50$, the list is discarded. We then shift the list by a random integer sampled from $[0, 50 - \max_i[l_i]]$. The list is discarded if it already exists in the dataset. We indicate the allowed $\delta$ range of these datasets in the superscript, such as $D_{\bar{\delta}\approx 4.8}^{\delta\in[2,8]}$.

It is important to recognize that any two datasets generated using different methods from the list described above, will not have equivalent distributions even if they share the same mean $\bar{\delta}$. This is because the spread and shape of the probability distributions of $\delta$ values will generally differ.

## 2.2 Measuring Development

In our developmental analysis, we study the evolution of attention head circuits alongside various measures. In this section, we describe these in more detail. As discussed in the seminal paper Vaswani et al. (2017), each attention head $h$ comprises various components that are learned during training, such as a query $W_Q^h$, key $W_K^h$, value $W_V^h$ and an output $W_O^h$ matrix. Using these matrices, we can decompose the calculation that an attention head performs into two largely independent circuits (Elhage et al., 2021)[2], namely the **Output-Value (OV) and the Query-Key (QK) circuits**, defined as

$$W_{OV}^h = W_E W_V^h W_O^h W_U \,, \qquad\qquad W_{QK}^h = W_E W_Q^h \left(W_K^h\right)^T W_E^T \,, \qquad\qquad (2)$$

where $W_E, W_U$ refer to the embedding and unembedding matrices, which change the basis to the more interpretable and intuitive token basis. The intuition behind how these circuits arise, is that the QK circuit determines the extent to which a particular query token should attend to a specific key token, whereas the OV circuit determines how the value of an attended token impacts the output.

We study a standard measure, the **validation loss** (referred to as simply loss[3], in the following), during training. We evaluate the validation loss on datasets generated in the same way as the datasets used for training, but with a different random seed. We have checked that there is no overlap between the sequences in the training sets and the validation sets. Additionally, we employ two other measures to capture **model complexity**:

- The **Local Learning Coefficient (LLC)** is a model-agnostic measure. It is always computed on a dataset that is distributed the same as the training distribution. The LLC was first introduced by Lau et al. (2023) for the purposes of evaluating model complexity, based on prior work in Singular Leaning Theory (SLT, Watanabe 2009). The LLC is computed by estimating the broadness of the local basin in the loss landscape. A larger LLC corresponds to a narrower basin and thereby a more complex model. The LLC is discussed in more detail in App. A.

- The **Circuit Rank** is defined as the sum of the matrix ranks of the OV and QK circuits. This measure is more specific to the model architecture compared to the previous two, as we directly employ the circuits in the definition. For an untrained model, the matrix rank is equal to the head dimension (48) for each of the circuits.

---

[2]Note that Elhage et al. (2021) uses a different convention for the weight matrices than `TransformerLens`, and we follow the convention of the latter. As an example, the $W_O^h$ matrices have dimension $[d_{\text{model}}, d_{\text{head}}]$ in Elhage et al. (2021), and $[d_{\text{head}}, d_{\text{model}}]$ in `TransformerLens`. This results in the formula for our OV and QK circuits differing from what can be found in Elhage et al. (2021).

[3]We don't observe a difference in the training and validation loss.

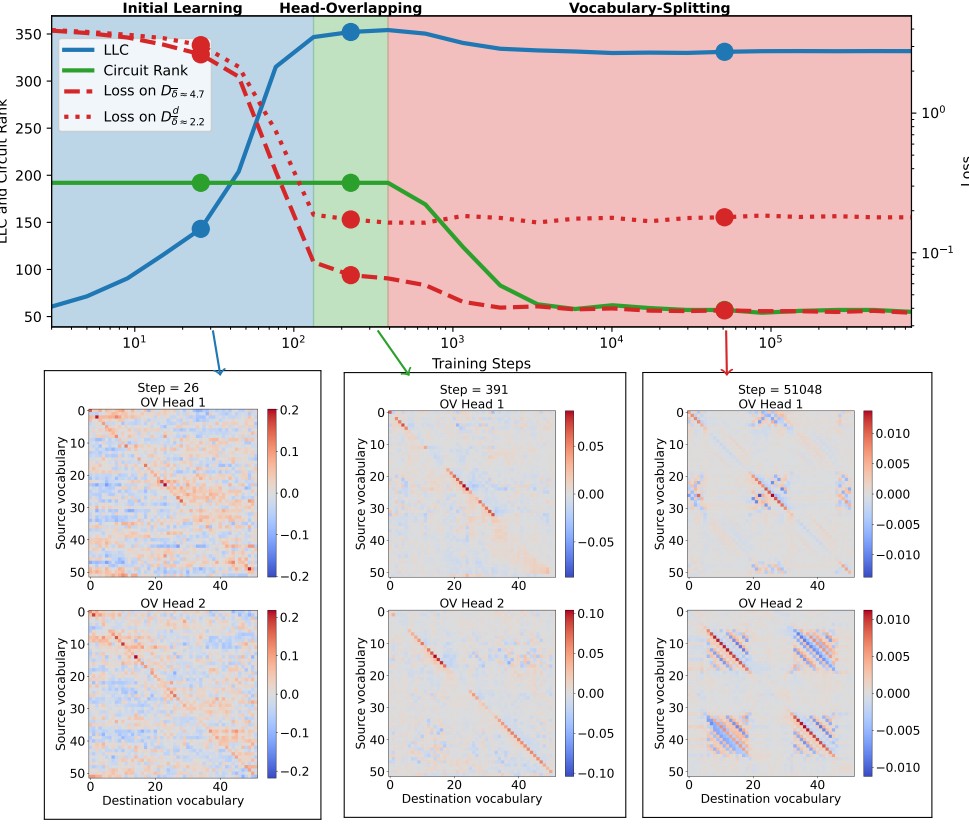

Figure 2: **Baseline 2-head model** trained on $D_{\bar{\delta}\approx4.7}$ undergoes three stages characterized by: rapid learning (left), heads copying partly overlapping vocabularies, as can be seen from the diagonal OV circuits (middle), and vocabulary-splitting head specialization with diagonal OV circuits covering contiguous regions (right). The loss on $D^d_{\bar{\delta}\approx2.2}$ measures out-of-distribution loss on lists with closer elements.

## 3 Results

In this section, we present the developmental stages and types of specialization the model goes through in Sec. 3.1, whereas in Sec. 3.2 we present results on what is driving the different specialization modes.

### 3.1 Developmental Stages and Specializations

We want to investigate how the model learns during training by looking at the evolution of the OV and QK circuits alongside various measures. In this section, we focus explicitly on the OV circuit for pedagogical reasons, and leave the QK circuits for App. B.2. In Figs. 2 and 3 we present the evolution of the baseline 2-head model during training on $D_{\bar{\delta}\approx4.7}$ and $D^d_{\bar{\delta}\approx2.2}$, respectively. The figures feature heatmaps of the OV circuits for the two attention heads, as well as an upper panel showing the LLC, the Circuit Rank and the loss evaluated on both $D_{\bar{\delta}\approx4.7}$ and $D^d_{\bar{\delta}\approx2.2}$. The distribution of $\delta$ in these datasets is shown in Fig. 4. The models go through the following stages:

1. For the first hundred steps, the models rapidly learn to sort, and we refer to this stage as **Initial Learning**. The loss decreases steeply on both $D_{\bar{\delta}\approx4.7}$ and on $D^d_{\bar{\delta}\approx2.2}$. The LLC on the other hand increases rapidly, whereas the Circuit Rank remains constant. During this stage the OV circuits start to form a diagonal structure as can be seen on the leftmost panel of Figs. 2 and 3.

2. As the loss and LLC flatten, the **Head-Overlapping** stage starts, characterized by fairly constant loss and LLC, and a constant Circuit Rank. On a circuit level, this stage is characterized by a clear

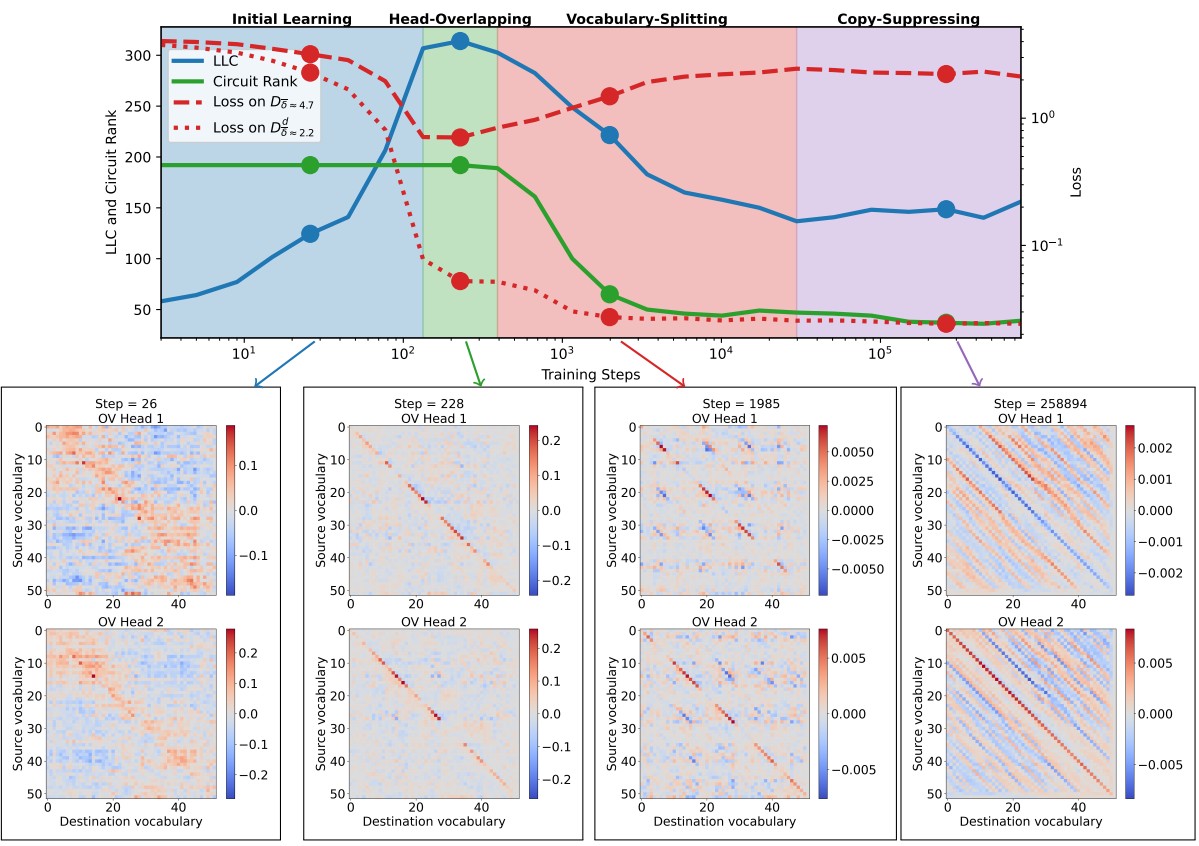

Figure 3: **2-head model** trained on $D^d_{\bar{\delta}\approx2.2}$ (the baseline model is trained on $D_{\bar{\delta}\approx4.7}$). Initially, it evolves similar to the baseline model(Fig. 2), but develops **copy-suppression** at the end of training. This is a gradual process, where the vocabulary covered by head 1 decreases before switching to copy-suppression.

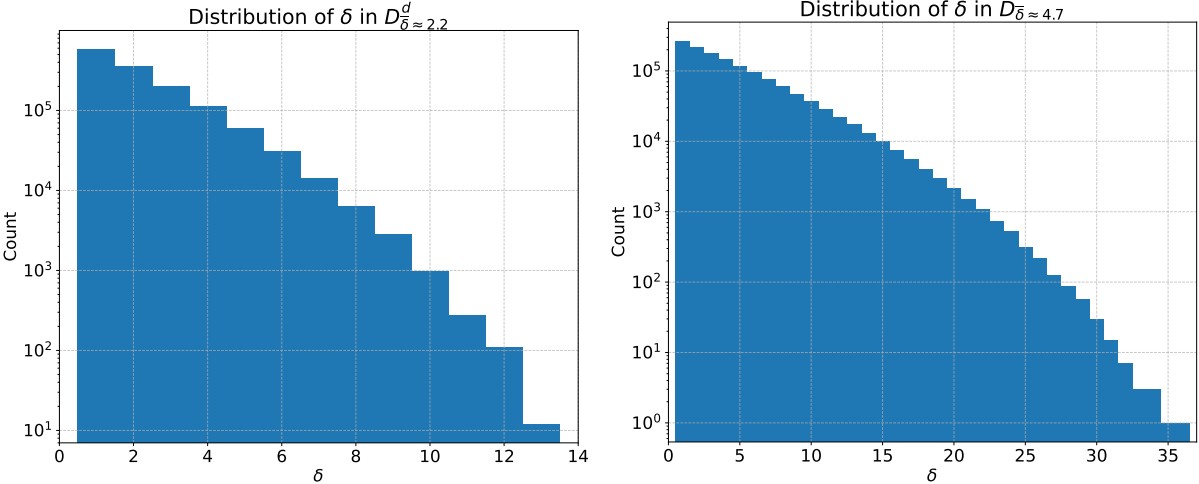

Figure 4: Distribution of $\delta$ (separation between neighbouring list elements) in $D^d_{\bar{\delta}\approx2.2}$ and $D_{\bar{\delta}\approx4.7}$.

diagonal structure in the OV, but the OV-diagonals of the two heads overlap, and they don't have the splitting of vocabulary yet. See second panel from the left of Figs. 2 and 3.

3. As the Circuit Rank starts to drop, the **Vocabulary-Splitting** stage starts, characterized by a drop in the in-distribution loss, but not in the out-of-distribution loss, and a decrease in the LLC. On the circuit level, this stage is characterized by a clear separation of the vocabulary between the heads, with OV circuits not overlapping (third panel from the left of Figs. 2 and 3). When training on $D_{\bar{\delta}\approx4.7}$ (Fig. 2), the LLC drops moderately before stabilizing together with the other measures, and the model remains stable until the end of training. If we instead train on $D^d_{\bar{\delta}\approx2.2}$ (Fig. 3), we note that the model develops a larger number of contiguous regions in the OV circuits. As opposed to the baseline model, the number of regions doesn't stabilize and the vocabulary regions in head 1 decrease throughout this stage, accompanied by a simultaneous decrease in the LLC and increase in the out-of-distribution loss on $D_{\bar{\delta}\approx4.7}$. The LLC continues decreasing until it reaches a minimum, where the final developmental stage starts.

4. When training on $D^d_{\bar{\delta}\approx2.2}$ (Fig. 3), the LLC reaches a minimum as the region size in head 1 can not decrease anymore, and a negative diagonal that covers the entire vocabulary range starts to form. Analogously, head 2 copies the entire vocabulary range. Since the QK circuits of both heads are similar (see right panel of Fig. 9 in the Appendix), we conclude that head 1 is doing **copy-suppression**, similar to what has previously been identified and discussed for `GPT-2` by McDougall et al. (2023) (see discussion in Sec. 4.3). To highlight this similarity, we termed this stage Copy-Suppression. It is the final stage that the model settles into when training on $D^d_{\bar{\delta}\approx2.2}$. We note that this model, compared to the baseline model that was trained on $D_{\bar{\delta}\approx4.7}$, shows a considerably larger loss on the dataset it was not trained on. This worse out-of-distribution performance and the larger drop in the LLC point towards the hypothesis that the model has learned a simpler solution that is more specialized to sorting lists with a small $\delta$.

In App. B, we **vary the number of heads and remove LN, WD or both**. If we only have 1 head, no head specialization can be present. When increasing the number of heads to 3-4, we find that two of the heads still specialize into vocabulary-splitting heads, whereas additional heads settle into full vocabulary copy-suppression or full vocabulary copying. Additionally, we find that removing WD leads to noisier circuits and weaker vocabulary-splitting, whereas removing LN causes the model to learn slower.

## 3.2 What drives Head Specializations?

In this section we investigate the role of the $\delta$ distribution, specifically, how this impacts head specializations and the size of the contiguous regions in the cases where vocabulary-splitting specialization is present. To this end, we train 2-head models on datasets with varying $\bar{\delta}$ (see Sec. 2.1 for how we vary this parameter). Before we present results, we shall take a small detour to introduce some relevant concepts related to the QK circuit.

In Fig. 5 we show the QK and OV circuits of the baseline 2-head model at the end of training. The dashed lines indicate the location of what we define to be the **active QK regions** for this model. The number in the top right corner of each active region corresponds to the region number in Tab. 1. We define the regions by first outlining the area above the diagonal in the QK circuit where the diagonal of the OV circuit is positive[4]. We then separate the active regions based on which OV region they have as input and output. As an example, region 2 (see top left panel of Fig. 5) has its input covered by the top left corner of the OV circuit of head 1 (see top right panel of Fig. 5), whereas its output is covered by the middle region of the OV circuit. This is important, as the attention pattern along a row at small vocabulary (top row of top left corner of Fig. 5) goes through regions 1, 2 and 4, so the attention pattern of region 2 and 4 is competing with region 1. Therefore, even though regions 2 and 3 share the same output (similar for regions 4, 5, 6), we distinguish between them by defining them as separate regions[5]. The regions that don't border the diagonal, i.e. regions 2, 4, 5 and 8 have a significantly lower prevalence[6] than the other regions, as can be seen in

---

[4]If more than one head has a positive OV diagonal for a given vocabulary, we define that vocabulary to belong to the head with the most positive OV diagonal. By definition, active regions of different heads don't overlap.

[5]Note that this definition is to some extent arbitrary, we could for instance have partitioned region 3 further. The important part here, is that we want to differentiate regions with low and high prevalence in the training dataset, since they exhibit qualitatively different patterns.

[6]The first list element in every list is excluded when computing the prevalence.

Table 1: Characteristics of the active regions in the QK circuits of the baseline model at the end of training, with region numbers defined in Fig. 5.

| Region number | Prevalence in training data | Mean region loss | Mean gradient $\nabla^{\mathcal{R}}$ |
|---|---|---|---|
| 1 | 3.3% | 0.039 | 8.2e-4 |
| 2 | 0.25% | 0.58 | 3.6e-4 |
| 3 | 28.0% | 0.037 | 3.6e-4 |
| 4 | 0 | - | -3.7e-4 |
| 5 | 0.24% | 0.30 | -3.3e-4 |
| 6 | 10.7% | 0.022 | 7.4e-4 |
| 7 | 27.0% | 0.031 | 4.3e-4 |
| 8 | 0.41% | 0.62 | 4.1e-6 |
| 9 | 30.1% | 0.033 | 2.6e-4 |

Tab. 1. This is because these regions are far from the diagonal and only contribute for large $\delta$, which is rare in the training dataset $D_{\bar{\delta} \approx 4.7}$ as shown in Fig. 4. We note that whereas the high-prevalence region behave as expected, with the QK circuit decreasing along rows from left to right, this is not always the case in the low-prevalence regions. This is not surprising, as the model is less incentivized to sort well in these regions, which is reflected in the higher mean loss as shown in Tab. 1.

To capture the degree to which the QK attends more to smaller tokens in a given active region, we define the **mean QK gradient** $\hat{\nabla}_{\mathrm{QK}}$. The gradient of an active region $\mathcal{R}$ containing $N_{\mathrm{rows}}$ rows starting at a column $j$ and ending at $k$, is calculated by taking the mean of the row gradients[7]:

$$\nabla^{\mathcal{R}} = \frac{1}{N_{\mathrm{rows}}} \sum_{i \in \mathcal{R}}^{N_{\mathrm{rows}}} \frac{w_{i,j} - w_{i,k}}{k - j}. \tag{3}$$

The resulting gradients $\nabla^{\mathcal{R}}$ are shown to the right in Tab. 1. To obtain the QK gradient of the entire model, we take the sum over the active region gradients $\nabla^{\mathcal{R}}$, weighed by the prevalence of the active region $\mathcal{R}$. The weighted sum is normalized by the combined mean weights across all the QK circuits of the model. A larger gradient means that the elements in the rows decrease faster from left to right, allowing the model to better distinguish neighboring list elements.

With all relevant quantities introduced, we return to the role of $\bar{\delta}$. Focusing on the $\delta$ distributions for which the model favors vocabulary-splitting, **we propose that a smaller $\bar{\delta}$ necessitates larger gradients within active regions of the QK circuits, driving region size down and number of regions up**. To build intuition for this hypothesis, assume that the elements in the rows of the QK matrix are linearly decreasing within the active regions, and that the gradient is the same for all the rows in the region. Furthermore, assume that in each region, the OV circuit is constant on the diagonal. The difference in attention paid to neighboring list elements within the same region is then only decided by the difference in the values of the corresponding row elements in the QK circuit. This difference in corresponding row elements is then proportional to the gradient of the row times the separation between the neighboring list elements, $\delta$. For example, if the sorted list contains the sequence [10, 12, 15], and the current token is 10, then head 2, region 7, does the sorting. The OV copies both 12 and 15 (and 10, but the QK is zero on the diagonal, suppressing 10), and by our assumption of constant diagonal it gives the same weight to both numbers. The attention paid to 12 and 15 is then decided by the difference between the 12th and 15th matrix element of row 10 in the QK matrix. If the elements are linearly decreasing with a slope equal to the gradient, then the difference between the two matrix elements is given by its separation (3 in our case) times the gradient. Relative to the overall scale of the weights in the matrix, the gradient can be larger the

---

[7]If the row begins at a diagonal element, we choose the next element to the right of the element on the diagonal as the leftmost element to calculate the gradient. Additionally, we divide by the length of the rows that *don't* start at the diagonal, to keep the normalization the same everywhere.

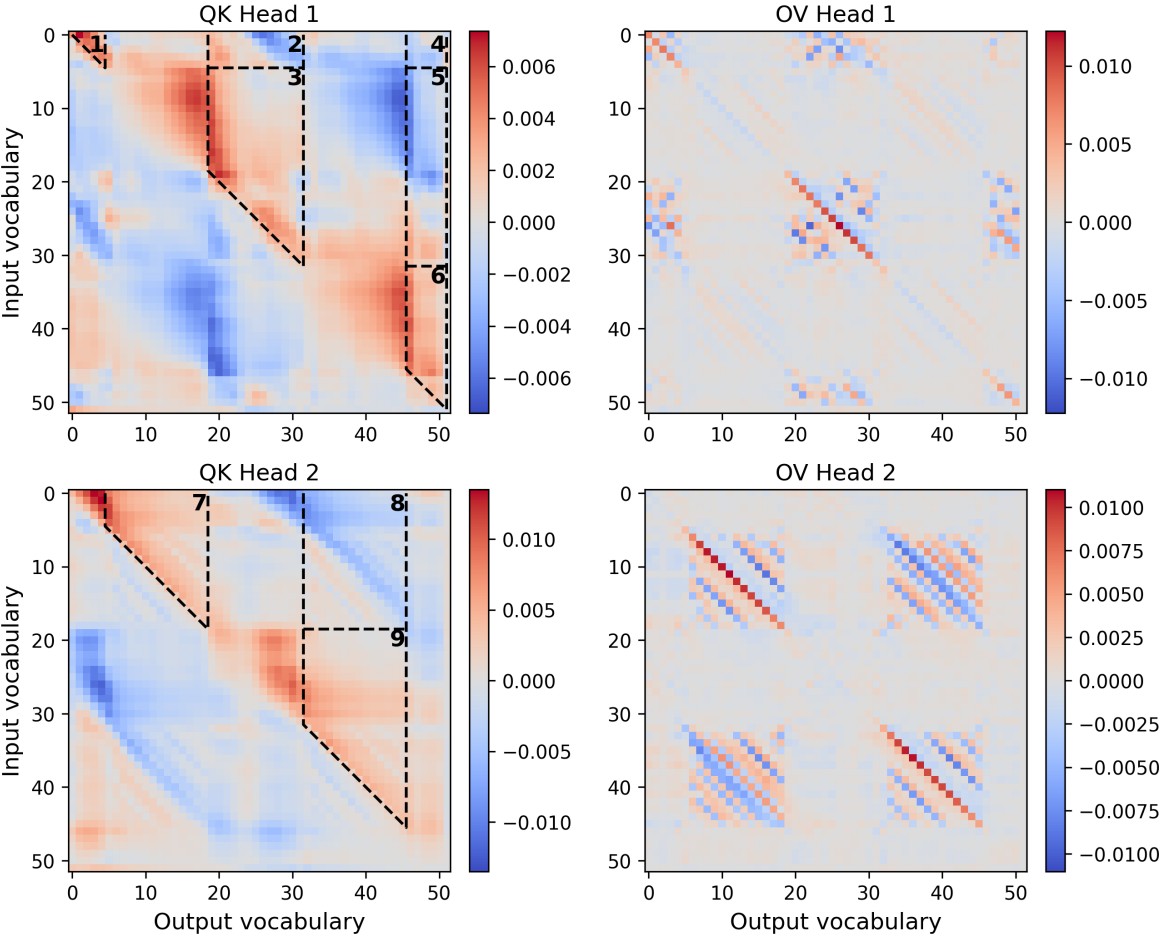

Figure 5: **Illustration of the active regions in the QK circuits** at the end of training for the baseline 2-head model trained on $D_{\bar{\delta}\approx 4.7}$. The attention pattern in regions close to the diagonal diminishes from left to right for each row, establishing a gradient. We hypothesize: the stronger the mean normalized QK gradients $\hat{\nabla}_{QK}$ of a model, the better the sorting of neighboring list elements with small separation.

Figure 6: **(Left)** The mean QK gradient of the active regions in the QK circuit $\hat{\nabla}_{QK}$ decreases proportionally with $\bar{\delta}$. To emphasize the flatness of the slope, we fit a linear line across all points with $\bar{\delta} \cdot \hat{\nabla}_{QK} = 0.004 \times \bar{\delta} + 0.468$. **(Right)** The mean region size increases with $\bar{\delta}$.

smaller the regions are. The hypothesis presented above in bold is supported by Fig. 6, where the model trains to a largely similar $\bar{\delta} \cdot \hat{\nabla}_{QK}$ across a large range of $\bar{\delta}$.

In App. C we summarize dataset variations for the 2-head models in Tab. 2 and discuss a selection of the setups in more detail. We find that for similar $\bar{\delta}$, small variance in $\delta$-values in the dataset leads to one head developing circuits orders of magnitude below the other head, a state we call **1-head sorting** due to the sub-leading head being "switched off". Increasing the variance causes the model to develop copy-suppression, and further increase causes the model to develop vocabulary-splitting.

In Fig. 16 we plot the weight norm of the sub-leading head relative to the weight norm of all heads, showing that the different specialization types cleanly separate into different ranges of relative weight norms. In some edge cases, such as when sorting lists with only three elements or when perturbing the training data, we observe a different kind of specialization: the OV circuits of both heads appear to be doing copying and copy-suppresion, but the corresponding QK circuits are different. Therefore we conclude that this is a different specialization mode.

## 4 Discussion

In this section we discuss the results presented in Sec. 3. In Sec. 4.1 we outline developmental stages recurring for many model setup variations, how we choose these stages and discuss an alternative approach to doing so. In Secs. 4.2-4.3 we discuss our insights regarding the head-specialization modes gained from varying the model setup.

### 4.1 Developmental Stages and How We Choose Them

**A common developmental stage sequence** that occurs in the baseline model and several variations of it, is illustrated in Figs. 2-3: 1) Initial Learning, characterized by rapidly decreasing loss and increasing LLC, 2) Head Overlapping, where both heads attend to and copy partly overlapping vocabularies and 3) a Head Specialization stage (one or a combination of vocabulary-splitting and copy-suppression). There are some exceptions: there is no head-overlapping stage when training without LN and with WD, and there is no head-specialization stage for the 1-head model (by definition).

**We choose the boundaries of a developmental stage** based on several factors, mostly related to significant changes in the measures and the patterns of the OV and QK circuits that are visible by eye. In the case of the loss and the complexity measures, a significant change constitutes variations in the steepness of the slope (Stage 1), reversal of the slope (Stage 2) or a local extrema (transition from vocabulary-splitting to copy-suppressing head specialization in Fig. 3). In the case of the circuits a significant change consists

in (relative) changes to the diagonal values of the OV circuit and changes to the active regions in the QK gradients $\hat{\nabla}_{\mathcal{R}}$. Our general heuristic consists in observing a combination of *some* of the changes mentioned above. An alternative approach to define stage boundaries is to always place them at local minima and maxima of relevant measures. This approach is followed by Hoogland et al. (2024). If we followed this approach and did not consult the OV and QK circuits, we would not define the intermediate head-overlap stage in our circuits, but we would still have the head-specialization stages.

**The developmental analysis approach helped us build the hypothesis regarding the role of $\delta$ in driving head specialization.** In particular the model in Fig. 3 provided an important hint for us to investigate why a model trained on a dataset with a smaller $\delta$ would develop vocabulary-splitting and abandon it during training in favor of copy-suppresion. We think this is a strong demonstration of the insights that can be gained by considering both developmental and mechanistic interpretability.

### 4.2 Vocabulary-Splitting is a Simpler State

**Vocabulary-splitting** head specialization is a recurrent feature of this model, even when removing LN, WD, both LN and WD or increasing the number of heads (for details, see Apps. B.3-B.7). Although the overall picture gets noisier, **vocabulary-splitting is robust with respect to these regularization techniques**. On the other hand, **vocabulary-splitting can be trained towards copy-suppression**, as we verified by further training the model from Fig. 2 on the dataset $D_{\frac{d}{\delta}\approx 2.2}^{d}$. In general, it is a **simpler model**, when compared to the preceding stage, where both heads attend to and copy overlapping vocabulary ranges, and its formation is always accompanied by a **drop in the LLC**. Importantly, the LLC decrease[8] is indicative of a solution that is both simpler *and* performs well on the task at hand, which distinguishes it from other model complexity proxies such as the Circuit Rank. This is exemplified in the 2-head model without LN (see Fig. 13 in the Appendix), where the Circuit Rank decreases significantly due to WD pushing the model to a simpler state, while the model only achieves 20% accuracy. The LLC on the other hand begins decreasing only later, when the transition to vocabulary-splitting occurs.

**The number of contiguous regions** that the vocabulary-splitting models settle into seems to be determined by the $\bar{\delta}$ in the dataset. As can be seen in Fig. 6, the product of the QK gradient $\hat{\nabla}_{QK}$ and $\bar{\delta}$ is largely model independent when varying $\bar{\delta}$ over an order of magnitude. This is because this product measures the ability to sort neighboring list elements, and guides the model development. Models can increase their gradients by decreasing their region sizes, thereby increasing the number of regions, and do so until the difference between the two most attended to tokens is large enough that softmax essentially sets the probability of the most attended to token to 1 and the others to 0.

The above argument relies on several approximations, and we don't expect it to hold exactly. Some key approximations going into the constant $\hat{\nabla}_{QK} \cdot \bar{\delta}$ are:

- The argument neglects variations in $\delta$, and approximates the lists in the training data as having a constant $\delta$ equal to the mean in the distribution.

- When computing the gradient of a row, we approximate the QK circuit as being linearly increasing/decreasing from left to right.

- By grouping rows together in regions, we approximate the prevalence of all rows in the region as being equal.

Despite these approximations, the model still seems to train to values of $\hat{\nabla}_{QK} \cdot \bar{\delta}$ which is fairly $\bar{\delta}$-independent, and to obtain region sizes which are approximately proportional to $\bar{\delta}$. We conclude that when varying $\bar{\delta}$, the model adjusts the region size, and thereby $\hat{\nabla}_{QK}$ so that $\bar{\delta} \cdot \hat{\nabla}_{QK}$ is "good enough" at sorting neighboring list elements in the dataset that further improvements will not have an important impact on the loss.

---

[8]We also observe a slight LLC decrease for the 1-head model, see App. B.1, where no vocabulary-splitting is possible.

### 4.3 Copy-Suppression helps calibrate the copying head

Copy-suppression is a second type of head specialization we encounter. As first defined by McDougall et al. (2023) for `GPT-2`, it is a mechanism consisting of three stages: copying, attention and suppression. **The copy-suppression we observe is visually similar to the one observed in `GPT-2`**, in the sense that we observe two heads paying attention to the same tokens (as can be seen from the similar QK circuits, right panel of Fig. 9), where one head directly suppresses the other copying head (see OV circuits in Fig. 3). Unlike vocabulary splitting, **copy-suppression is not easily trained towards a different head specialization**. Training the model from Fig. 3 further on the default dataset $D_{\bar{\delta}\approx4.7}$ doesn't change the model away from this mode. We believe this to be due to two (somewhat related) factors:

1. The copy-suppression has a considerably lower LLC than the vocabulary splitting model, as can be seen in by comparing the end-of-training LLCs shown in figs. 2 and 3. We expect a broader loss-landscape basin to be harder to escape.

2. The vocabulary-splitting model can be smoothly perturbed towards copy-suppression by gradually reducing the size of the vocabulary regions covered by the head initially covering the least of the vocabulary. As the regions covered by the least dominant head shrinks, the other head gradually takes over these regions before the shrinking vocabulary vanishes completely and the least dominant head switches to doing copy-suppression. The same is not true when going from copy-suppression to vocabulary splitting.

Next, we investigate **the functional role** of copy suppression. In McDougall et al. (2023) the authors investigate multiple hypotheses for the role of copy-suppression in `GPT-2`, such as acting as a calibrating head for the copying head to reduce model overconfidence, and contributing to the self-repair phenomenon (McGrath et al., 2023), where neural network components compensate for ablations or perturbations of parts of the network. They find support for both hypotheses, and temporarily conclude that copy-suppression is a form of self-repair, but that doesn't exclude the second hypothesis. Here we should note that there are some important architectural differences with respect to our setup. In `GPT-2`, the copy-suppressing heads come after the copying heads, and get the copied token as part of its input. Our copy-suppressing heads work in parallel with the copying heads. This might be an important difference, as the case McDougall et al. (2023) makes for copy-suppression correcting an error made by a previous head is harder to make in our case when the copy-suppressing head cannot "see" the output of the other heads. Furthermore, the copy-suppressing circuit in `GPT-2` includes an MLP, whereas we don't.

In our model, **copy-suppression seems to play a supportive role** in slightly modifying the dominant copying attention head. We find some evidence that **it is aiding in increasing model confidence in predictions**, instead of decreasing it. To reach these conclusions, we conduct two simple experiments:

1. We ablate each head in succession for the model from Fig. 3, by setting the ablated heads output to the mean output across the dataset, see Fig. 10 in the Appendix. We find that **ablating the copy-suppressing head has no impact on the accuracy of the model, but it does increase the loss slightly**, with the copying head clearly dominating the performance of this model.

2. We calculate the Shannon Entropy of the model, which is high when the logprob distribution is flat and low if it is peaked, before and after ablating each head. In both cases, this measure increases after ablation.

Putting these experiments together, we conclude that the copy-suppressing head slightly increases model confidence by increasing the "correct logits" and making the logit distribution more peaked. This way, it is a preferred solution to completely turning this head off, since this gives a slightly lower loss. This functional role of copy-suppression in our model is in partial disagreement with one of the hypotheses discussed for the related mode in `GPT-2`.

**The transition between copy-suppression, vocabulary splitting and yet another mode we call 1-head sorting gives further insight in understanding this model.** In Fig. 3 we observe that head 1

gradually covers a smaller and smaller vocabulary range, before it switches over to doing copy-suppression. We believe that this happens because $D_{\bar{\delta}\approx 2.2}^{d}$ is comparatively simple, in the sense that it has a smaller variance than $D_{\bar{\delta}\approx 4.7}$ (see Fig. 4). Due to the dataset simplicity, the model does not benefit sufficiently from having two heads, and WD pushes the weights of head 1 down. It is, however, limited how small the weights of a head can be (relative to those in the other head) if they are to dominate in a vocabulary range. The model therefore switches to having both heads cover the entire vocabulary range, where head 1 makes a small adjustment to the sorting of head 2 across the entire vocabulary range. For even simpler datasets, WD pushes the weights of the sub-leading head down further, producing circuits orders of magnitude below those in the leading head. We refer to this state as **1-head sorting**. In Fig. 16, we show the weight norm of the sub-dominant head relative to the combined weight norm of all the heads, showing that the different modes of specialization (vocabulary-splitting, copy-suppression and 1-head sorting) cleanly separate into different ranges of sub-leading head weight norm. Which datasets produce which specialization can be found in Tab. 2 together with the mean and variance of $\delta$. The table shows that **for similar $\bar{\delta}$, the models develop 1-head sorting, copy-suppression and vocabulary-splitting in order of increasing $\delta$ variance, respectively**.

## 5   Related Work

Stagewise development in artificial neural networks is not a new field of study, see e.g. Raijmakers et al. (1996). Hoogland et al. (2024) found developmental stages, including a drop in the LLC corresponding to model simplification, when training a transformer on linear regression. The LLC evolution of non-transformer toy models has previously been studied by Panickssery & Vaintrob (2023) and Chen et al. (2023). Without using the LLC, Chen et al. (2024) studied developmental stages in BERT. Bagiński & Kolly (2023) and McDougall (2023b) studied algorithmic transformers trained on list sorting, Nanda et al. (2023) reverse-engineered an MLP trained on modular addition, and Power et al. (2022) trained a transformer on modular addition to study grokking. In this paper, we find copy-suppression, previously observed by McDougall et al. (2023).

Previous research on the universality hypothesis has found evidence of specialized components called **induction heads** (Olsson et al., 2022) that help predict repeating sequences, a pattern previously found in larger language models.

Study of toy models in mechanistic interpretability has also been carried out on additional networks trained on modular arithmetic (Chughtai et al., 2023; Stander et al., 2023), and has been used to demonstrate the presence of super-position in neural networks (Elhage et al., 2022).

## 6   Limitations

Our study is done on a toy model, and one should be careful to generalize our findings to larger transformers. Additionally, our interpretation of the functionality of the circuits is approximate, and we expect there is probably more going on in the model.

In our study of the impact of the $\delta$ distribution, we have not done intervention studies increasing or decreasing the gradients to confirm that they shift the delta range the model sorts the best at. We also expect the location of threshold effects we observe to be sensitive to the strength of the WD. We have not checked this.

We have tested three different random seeds for the baseline 2-head model and found that the number of regions can fluctuate by up to 1 vocabulary region, but other results remain intact. We have not systematically checked for seed dependence, and the outcome of different seeds can be seen in Fig. 16. We don't expect our main findings to be seed-dependent.

The LLC is only defined at a local minimum, which models during training never are at in practice. Lau et al. (2023) argues that the LLC value is not trustworthy, but that the relative ordering of LLCs at different stages of training is. The LLC hyperparameter selection is not an exact science, and in this paper we used the heuristics of seeking parameter space in which the LLC is locally hyperparameter independent.

## 7 Conclusion

We present a theoretical study of an attention-only list-sorting algorithmic transformer, extending earlier work by McDougall (2023b) on understanding how these models function internally and how learned solutions arise and form during training. Our developmental analysis reveals that **attention heads naturally organize themselves into different specialized roles** at the end of training - they either split up different numerical ranges between them (**vocabulary-splitting**) or develop a system where one head handles copying and sorting numbers while another head fine-tunes those predictions (**copy-suppression**).

The **developmental approach** gave us an important initial hint in pursuing the transition between the head specializations. This led us to **discover features of the training data that strongly influence which type of specialization emerges at the end of training**, specifically, the mean gap $\bar{\delta}$ between neighboring (sorted) list numbers and its variance. For example, when adjacent sorted numbers tend to have larger gaps between them with high variance (more diverse dataset), the model develops vocabulary-splitting. However, when gaps are smaller with less variance (less diverse dataset), the model instead develops copy-suppression. If the variance is reduced beyond some threshold, the copy-suppressing head is ultimately switched off.

**These findings help us extend the original interpretation of the model** presented by McDougall (2023b). Specifically, for vocabulary-splitting, we hypothesize that small $\bar{\delta}$ necessitates the attention pattern in the QK circuit to distinguish between neighboring vocabulary elements more precisely, thus increasing the difference of their QK matrix values. With some approximations, this should lead to a constant product $\bar{\delta} \cdot \hat{\nabla}_{QK}$, which we confirm by varying $\bar{\delta}$.

We show that models develop simpler solutions naturally during training, even without explicit regularization methods thought to promote simplicity, like weight-decay (WD). This is the case for the **vocabulary-splitting** head specialization, which is simpler than the preceding stage during training where different heads attend to overlapping number ranges. It is **an important concrete example suggesting that neural networks have an inductive bias towards simpler and more general solutions that can give a better loss on diverse training datasets**, rather than maintaining more complex overlapping functionality between components.

**Copy-suppression** constitutes an intermediate state between vocabulary-splitting and switching off one head, where training on a dataset with an intermediate $\delta$ variance is still diverse enough to favor *not* switching off a somewhat superfluous head, but rather employ it towards some other purpose. We demonstrate how copy-suppression works in our model, showing that it serves as a calibration mechanism to increase model confidence - a contrast to previous findings in `GPT-2` McDougall et al. (2023), where copy-suppression tends to reduce overconfidence. This is probably due to our simpler toy-model having a 100% accuracy, making an increase in confidence more beneficial than a decrease. This difference helps us understand that copy-suppression more broadly acts to calibrate the model's predictions based on accuracy. **These findings elucidate the broader role of copy-suppression as a calibration mechanism for model predictions based on accuracy, contextualizing the relevance of our toy-model for more complex models and providing nuanced insights into the functional role of copy-suppression across different architectures.**

The findings listed above open up several promising directions for future research. One key area would be studying how these developmental patterns appear in more complex models that have already been partially interpreted. This could help us understand why models develop certain internal structures and how different components coordinate with each other. Another important direction would be leveraging these insights to develop training techniques that encourage models to develop more interpretable or desirable internal organizations.

Our findings demonstrate that it's possible to build detailed, mechanistic understanding of how neural networks organize themselves across different training distributions. This type of understanding could be crucial for developing more reliable and controllable AI systems, as it gives us insight into not just what these models do, but how and why they develop particular ways of solving problems.

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

## A  Singular Learning Theory and the Local Learning Coefficient

Our main tool for studying model development is the Local Learning Coefficient (LLC), a theoretically well-motivated measure of model complexity based on the learning coefficient from Singular Learning Theory (SLT) (Watanabe, 2009). The LLC is defined in Definition 1 of Lau et al. (2023) as a unique rational number such that asymptotically as $\epsilon \to 0$, the volume of model parameters $\omega$ around a minimum $\omega^*$ in which $L(\omega) - L(\omega^*) < \epsilon$ can be written as

$$V(\epsilon) = c\epsilon^{\text{LLC}(\omega^*)}(-\log \epsilon)^{m(\omega^*)-1} + o\left(\epsilon^{\text{LLC}(\omega^*)}(-\log \epsilon)^{m(\omega^*)-1}\right), \tag{4}$$

where the positive integer $m(\omega^*)$ is the local multiplicity and $c > 0$ is a constant.

The LLC is a measure of the degeneracy of the loss landscape near a model's parameters $w^*$, where a lower LLC indicates a more degenerate and less complex model. Given an empirical loss $\ell_n(w)$ over parameters $w$ computed with a batch size of $n$, we calculate the LLC estimate at a local minimum $w^*$ similar to Hoogland et al. (2024) and Lau et al. (2023):

$$n\beta\left[\mathbb{E}_{w|w^*,\gamma}^{\beta}[\ell_n(w)] - \ell_n(w^*)\right],$$

where $\mathbb{E}_{w|w^*,\gamma}^{\beta}$ denotes the expectation with respect to a tempered posterior distribution centered at $w^*$ and $\beta = 1/\ln(n)$ is the inverse temperature. $\gamma$ controls the localization around $w^*$, and is set so that the sampled parameter space is sufficiently large that the volume of the local minimum is captured, but not so large that nearby local minima are included. We set $\gamma$ so that the LLC is locally $\gamma$-invariant. The sampling is done with Stochastic Gradient Langevin Dynamics (SGLD).

The LLC is calculated using the `DevInterp v.0.2.2` software package (van Wingerden et al., 2024). The hyper-parameters vary with the setup, and are found by performing parameter scans, where we look for regions of parameter space where the LLC is hyper-parameter independent.

## B  Varying the Model Architecture and Training

In this subsection, we study the impact of varying the model architecture and training such as the number of attention heads, and the use of LN and WD.

### B.1  1-Head Model

In Fig. 7 we show the development of a 1-head model trained on our list-sorting task. Note that the attention head has dimension 48 like the heads in the 2-head models. Like the baseline 2-head model, this model undergoes three distinct stages, with a stage of initial learning with a rapidly decreasing loss and rapidly increasing LLC, an intermediate stage where the loss and LLC is fairly constant with a decrease

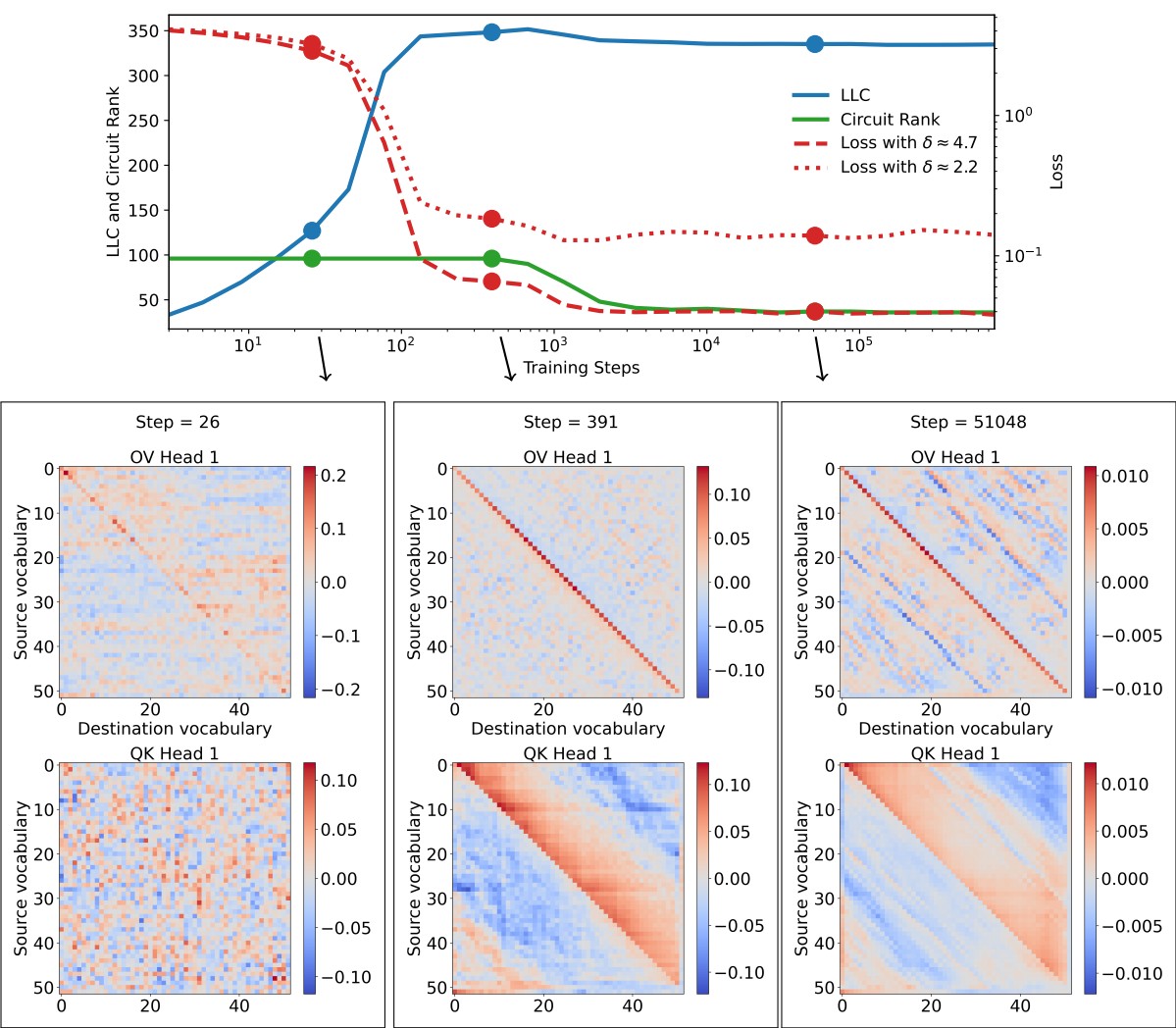

Figure 7: **1-head model** trained on $D_{\bar{\delta}\approx4.7}$ undergoes three stages characterized by: rapid learning (left), QK and OV circuits develop the expected patterns (middle) and off-diagonal patterns appearing in the OV circuit (right). The loss is evaluated on $D_{\bar{\delta}\approx4.7}$ and $D^d_{\delta\approx2.2}$.

in the Circuit Rank and the Loss towards the end of this stage, and finally a stable stage characterised by off-diagonal stripes in the OV circuit. As there is only one head, the model can not undergo vocabulary-splitting, and only has a slight reduction in the LLC as the Circuit Rank drops and the off diagonal patterns form.

The LLC has been calculated with inverse temperature $n\beta = 512/\ln 512 \approx 82$, step size $\epsilon = 10^{-4}$, localization term $\gamma = 32$, $n_{\text{chains}} = 4$ and $n_{\text{draws}} = n_{\text{burnin}} = 2000$.

## B.2 Baseline 2-Head Model

In Fig. 8 we show both the QK and OV circuits of the developmental stages the baseline 2-head model undergo. We observe that the QK circuit becomes more regional in the vocabulary-splitting stage, reflecting the specialization in the OV circuit.

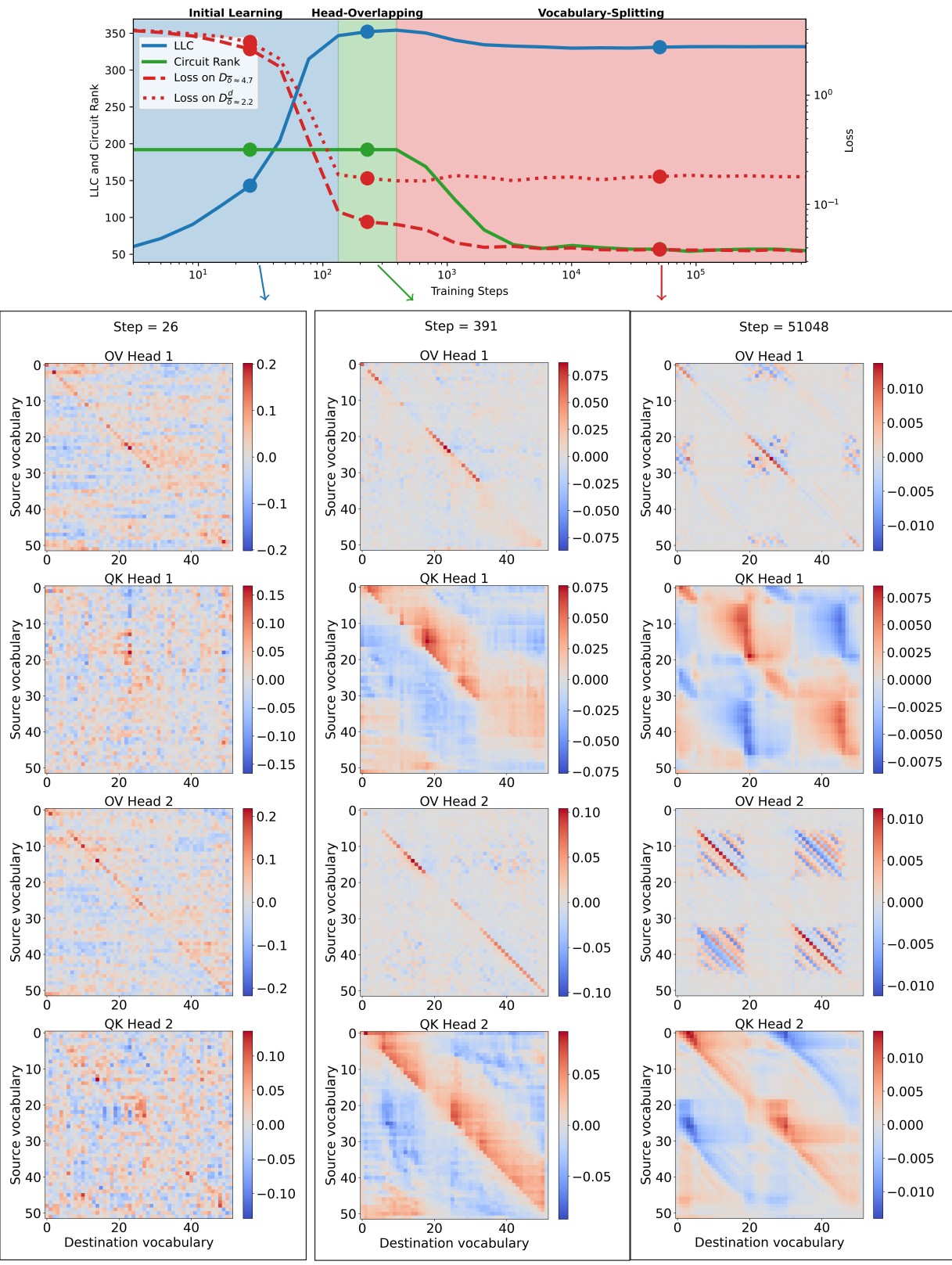

Figure 8: **Both OV and QK circuits of the baseline 2-head model** trained on $D_{\bar{\delta}\approx4.7}$ during the developmental stages.

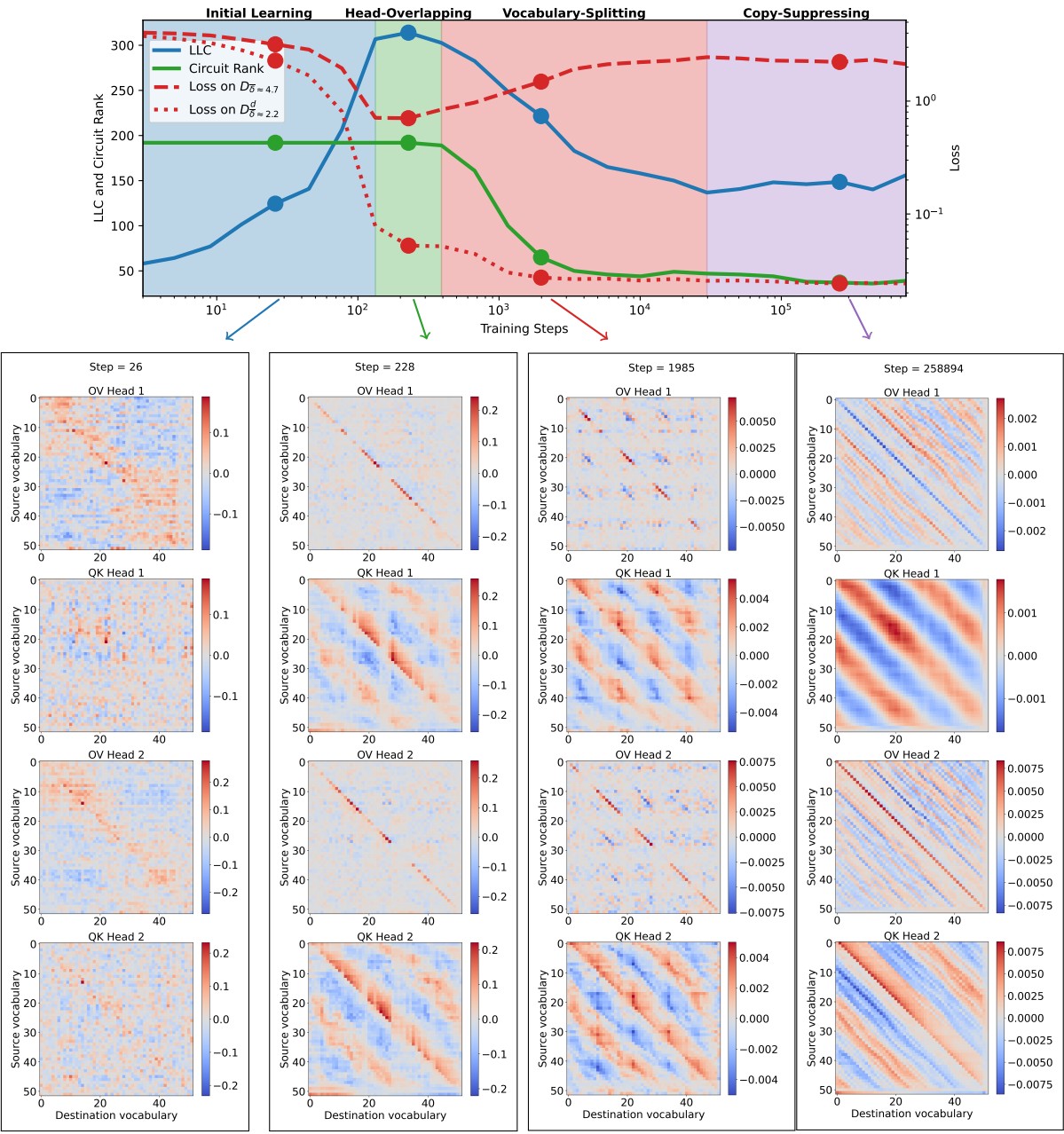

Figure 9: **Both OV and QK circuits of the baseline 2-head model** trained on $D^d_{\overline{\delta} \approx 2.2}$ during the developmental stages.

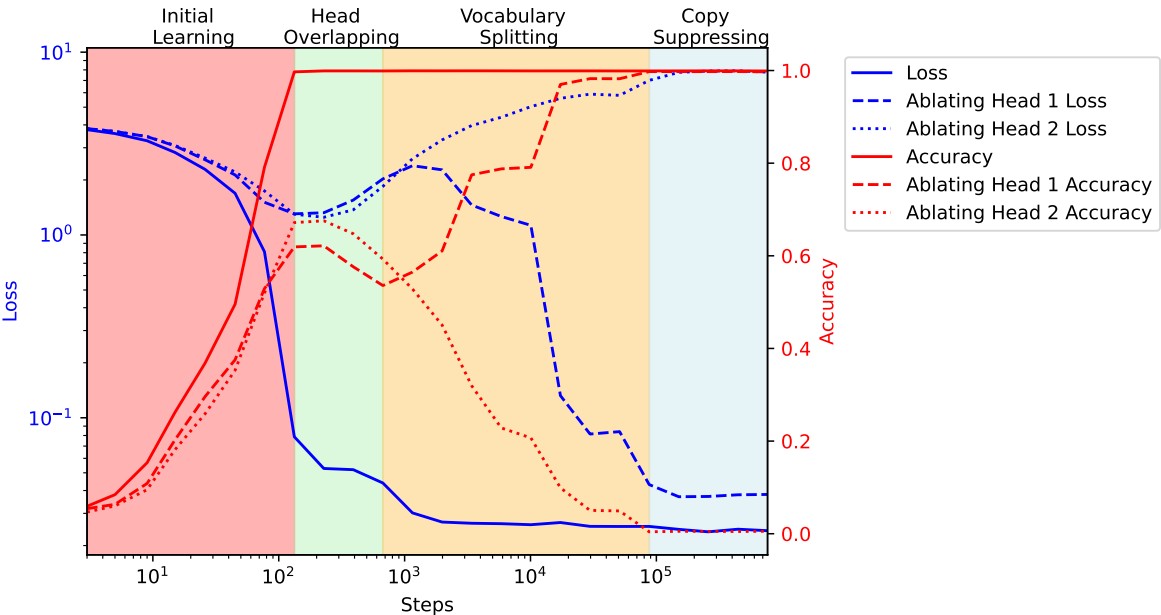

Figure 10: Head ablation loss and accuracy of the model trained on $D_{\bar{\delta}\approx2.2}^d$.

In Fig. 9 we show both the QK and OV circuits of the developmental stages that the model undergoes when trained on $D_{\bar{\delta}\approx2.2}^d$. We observe in the 3rd column that the QK displays a periodic pattern matching that of the OV, and that in the 4th column the QKs of both heads are similar across the vocabulary.

In Fig. 10 we show the loss and accuracy when ablating a head of the model showed in Fig. 9. We see that after copy suppression has started, ablating head 1 does not change the accuracy, but it does increase the loss. Thereby, we can conclude that head 1 is adjusting the output of head 2 to increase the confidence, thereby reducing the loss.

The LLC of the baseline 2-head model has been calculated with inverse temperature $n\beta = 512/\ln 512 \approx 82$, step size $\epsilon = 3 \times 10^{-5}$, localization term $\gamma = 56$, $n_{\text{chains}} = 4$ and $n_{\text{draws}} = n_{\text{burnin}} = 30000$. The hardware used to calculate the LLC is NVIDIA RTX-4090. When training and calculating the LLC on $D_{\bar{\delta}\approx2.2}^d$ we use $n\beta = 512/\ln 512 \approx 82$, $\epsilon = 5 \times 10^{-5}$, $\gamma = 32$, $n_{\text{chains}} = 4$ and $n_{\text{draws}} = n_{\text{burnin}} = 5000$.

### B.3 3-Head Model

As shown in the first row of Fig. 11, the **3-head model** (trained with head dimension of 48) features a loss that decreases rapidly until step 133 (top left of Fig. 11), where all heads attend to and copy overlapping vocabulary regions. At peak LLC (top right of Fig. 11) we see first signs of vocabulary-splitting head specialization. As the LLC drops, the overlap between their vocabulary regions decreases, resulting in contiguous regions split across three heads, with head 3 covering only a small region and starting to show signs of a negative diagonal (bottom left Fig. 11). The QK circuits also display differentiated patterns, which upon closer inspection match the active vocabulary regions of the OV circuits. So far, the developmental stages of this model, match those of the baseline 2-head model.

As the evolution continues, around training step 5859 (not shown) the OV circuit of head 3 specializes to a negative diagonal, seemingly suppressing the contributions from the other two heads, which behave like in the baseline 2-head model. We identify the state of head 3 to be copy-suppression. As the transition occurs, the QK circuit of head 3 also switches to uniform diagonal patterns, not differentiating any vocabulary regions anymore. This transition corresponds to a drop in the out-of-distribution loss on $D_{\bar{\delta}\approx2.2}^d$, and is not captured by any of the other measures.

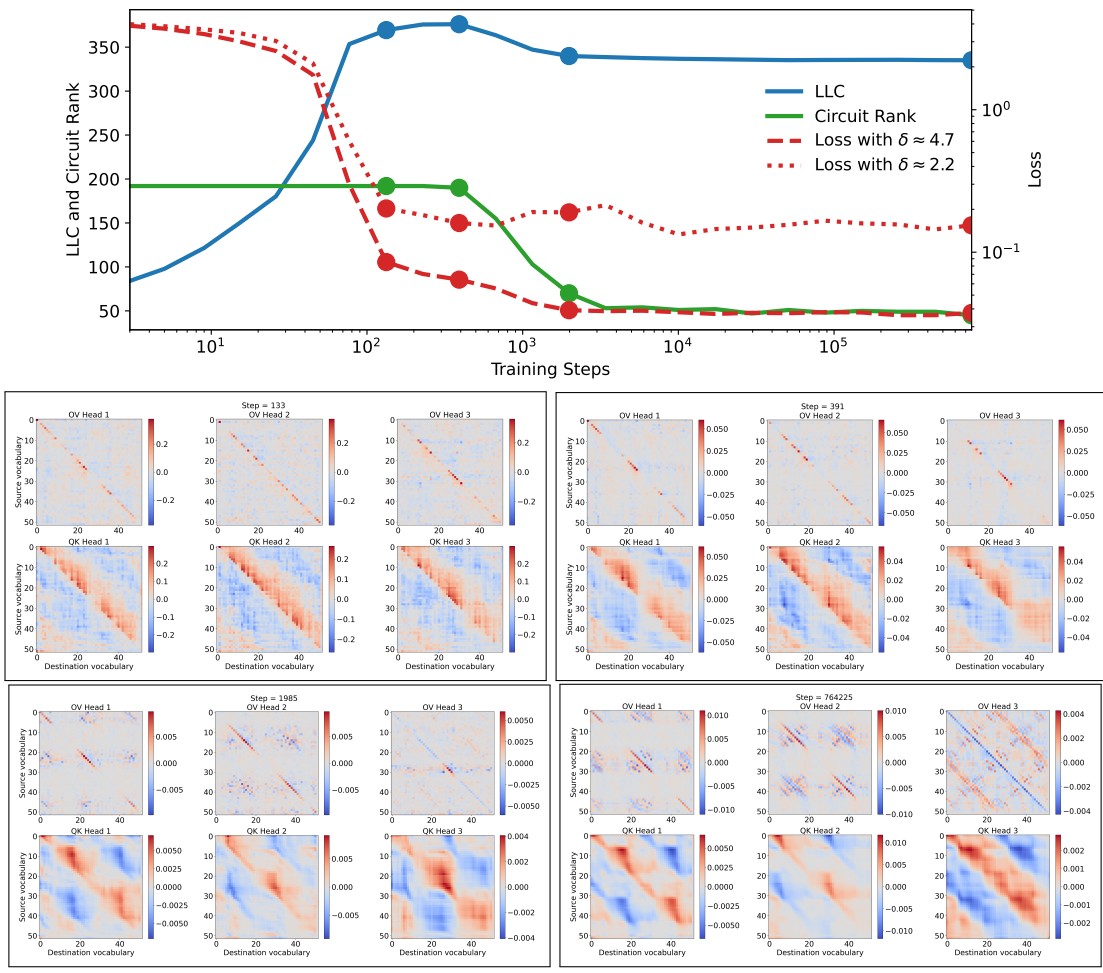

Figure 11: **3-head model** trained on $D_{\bar{\delta}\approx 4.7}$. As the model learns how to sort (top left), at LLC peak (top right), three-way vocabulary-splitting after LLC decrease (bottom left) and head 3 performing copy-suppression (bottom right). The loss is evaluated on $D_{\bar{\delta}\approx 4.7}$ and $D^d_{\bar{\delta}\approx 2.2}$.

The LLC has been calculated with inverse temperature $n\beta = 512/\ln 512 \approx 82$, step size $\epsilon = 10^{-4}$, localization term $\gamma = 32$, $n_{\text{chains}} = 4$ and $n_{\text{draws}} = n_{\text{burnin}} = 60000$.

### B.4 4-Head Model

Similar to the other models, the **4-head model** (trained with head dimension of 48) also starts with a sharp decrease in the loss until step 133 (1st row of Fig. 12). As the LLC decreases, heads begin to specialize with concurrent vocabulary-splitting and copy-suppression appearing in heads 1,3,4 and head 2 respectively (2nd row of Fig. 12). We note that this happens as the out-of-distribution loss on $D^d_{\bar{\delta}\approx 2.2}$ decreases. The vocabulary regions are split unevenly, with head 4 covering only a very small region of the vocabulary.

This changes later in the training, after around 87k training steps (3rd row of Fig. 12), with heads 3 and 4 now copying similar vocabulary regions and displaying differentiated attention patterns in the QK circuits. Directly after this, at step 150k, head 3 grows to attend and copy the entire vocabulary range. It seems to transition to do the bulk of the sorting with heads 1 and 4 doing minor adjustments. This last transition is captured by a small drop in the LLC. The model remains largely unchanged after this point, until the end of training (4th row of Fig. 12), as is seen from the measures remaining fairly constant.

The LLC has been calculated with inverse temperature $n\beta = 512/\ln 512 \approx 82$, step size $\epsilon = 10^{-4}$, localization term $\gamma = 32$, $n_{\text{chains}} = 4$ and $n_{\text{draws}} = n_{\text{burnin}} = 2000$.

### B.5 Baseline 2-Head Model without LN

Removing LN from the baseline 2-head model causes a dramatic change to the training dynamics, as shown in Fig. 13. Early in training, at steps 71-348 (top row) the model goes through a transition in which The Circuit Rank drops dramatically. During this transition, the circuits of the model form a very regular dipole-like pattern.

This dipole-like pattern starts breaking at steps 18298-27194 (middle row) as the LLC peaks, with a formation of the stripe-like patterns parallel to the diagonal in the OV circuit. The QK circuits cover the regions determined by the OV circuit, similar to what we have seen in the other models. This structure formation stabilizes as the LLC drops (bottom row), which is also tracked by a strong decrease in the loss on both datasets. The model never reaches 100% accuracy on list sorting, and the loss does not flat-line until step 60060, after which all the measures are stable. The LLC seems to capture the development of this model very well.

The LLC has been calculated with inverse temperature $n\beta = 26$, step size $\epsilon = 10^{-6}$, localization term $\gamma = 32$, $n_{\text{chains}} = 3$ and $n_{\text{draws}} = n_{\text{burnin}} = 100000$.

### B.6 Baseline 2-Head Model without WD

As seen in Fig. 14, the model without WD learns to sort with the diagonal OV and positive band above the diagonal in the QK at step 133. Compared to the baseline model, the OV and QK circuits seem more noisy, and there is no drop in the Circuit Rank. The LLC still has a large drop between steps 1985 and 10066 during which the heads specialize into splitting the vocabulary, and the loss decreases further. This specialization is clearer for tokens smaller than 20 in the QK and OV circuits, less so for larger vocabulary tokens. Unsurprisingly, we note that this model achieves the lowest loss of any of the models we train. With an out-of-distribution loss of about 0.01 on $D_{\bar{\delta}\approx 0.22}$, this model achieves a better loss also on this dataset.

The LLC has been calculated with inverse temperature $n\beta = 512/\ln 512 \approx 82$, step size $\epsilon = 3 \times 10^{-6}$, localization term $\gamma = 56$, $n_{\text{chains}} = 4$ and $n_{\text{draws}} = n_{\text{burnin}} = 65000$.

### B.7 Baseline 2-Head Model without LN and WD

Fig. 15 shows the evolution of our measures and the circuits for the baseline 2-head model without both LN and WD. The model seems to go via dipole-like circuits around step 45, very similar to step 71 of the baseline model without LN (compare the top left panels of Figs. 13 and 15). Instead of going via the low

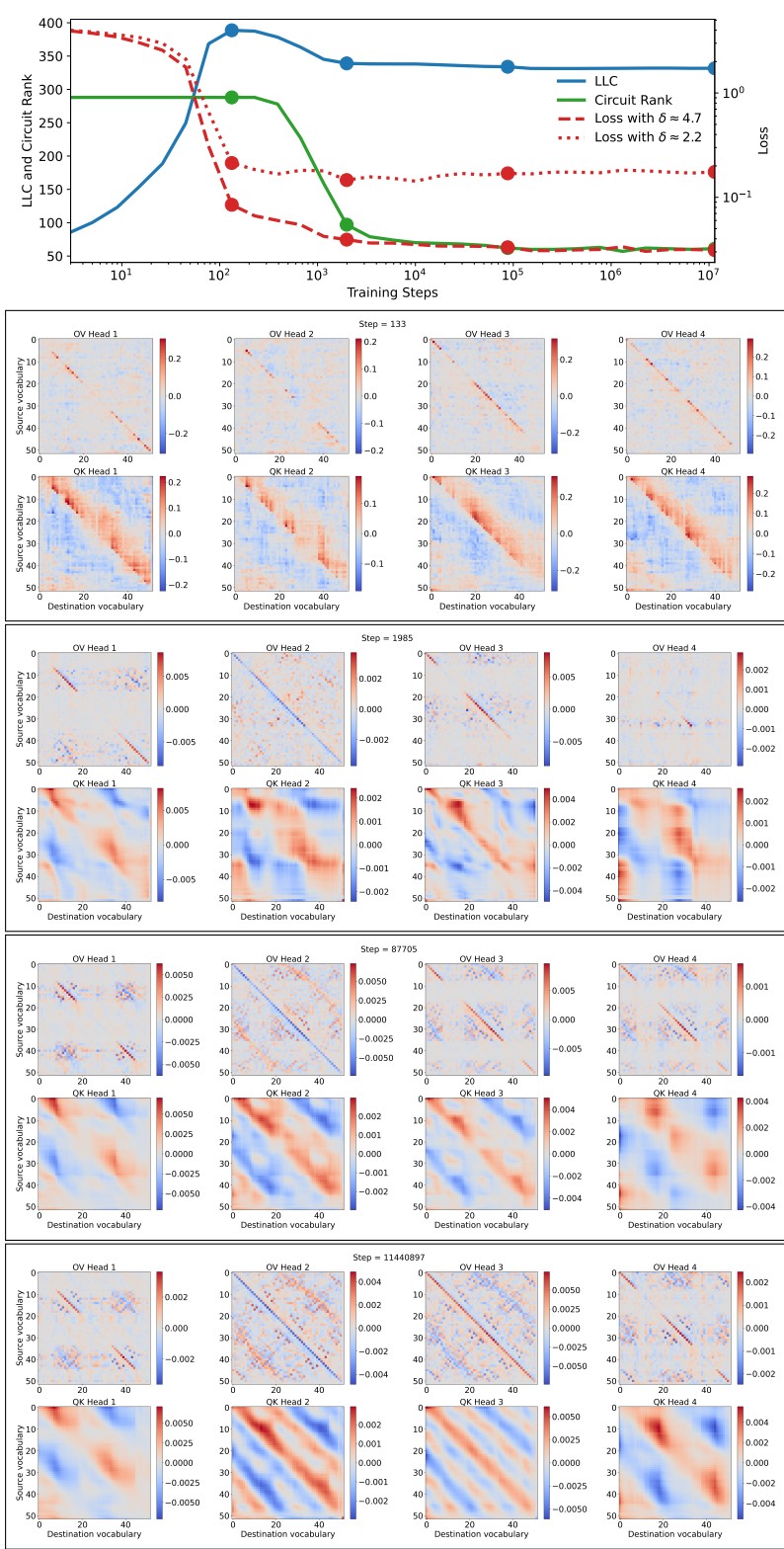

Figure 12: **4-head model** trained on $D_{\bar{\delta}\approx4.7}$. As the model learns how to sort (1st row), as the LLC decreases and heads specialize differently (2nd row), as heads 3 and 4 cover the same vocabulary regions (3rd row), as head 3 covers the entire range (4th row), and at the end of training (5th row). The loss is evaluated on $D_{\bar{\delta}\approx4.7}$ and $D^d_{\bar{\delta}\approx2.2}$.

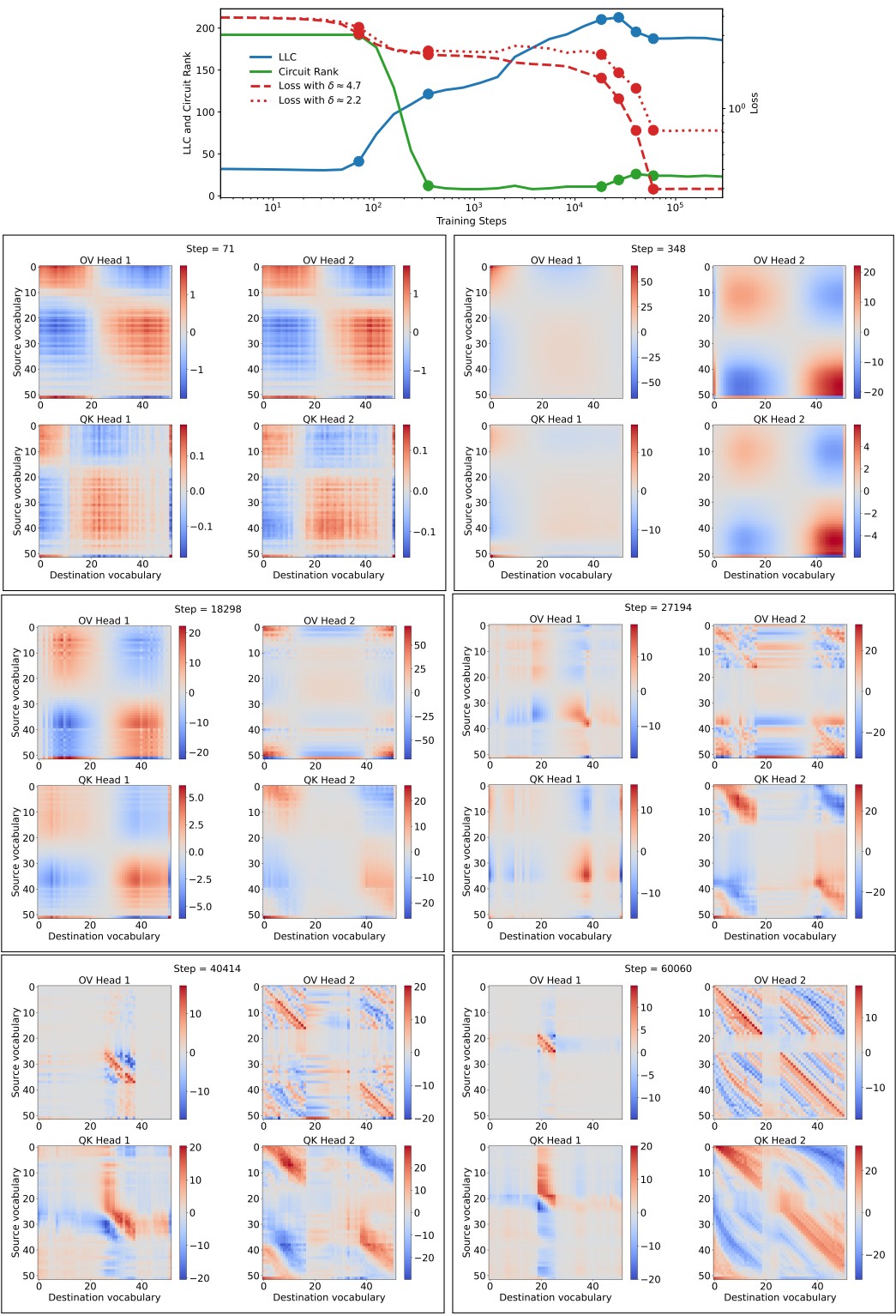

Figure 13: **Baseline 2-head model trained without LN** trained on $D_{\bar{\delta}\approx4.7}$. As the model simplifies but performs poorly (1st row), as relevant structure develops and performance improves rapidly (2nd row), as vocabulary-splitting appears before and after LLC decrease (3rd row). The loss is evaluated on $D_{\bar{\delta}\approx4.7}$ and $D^d_{\bar{\delta}\approx2.2}$.

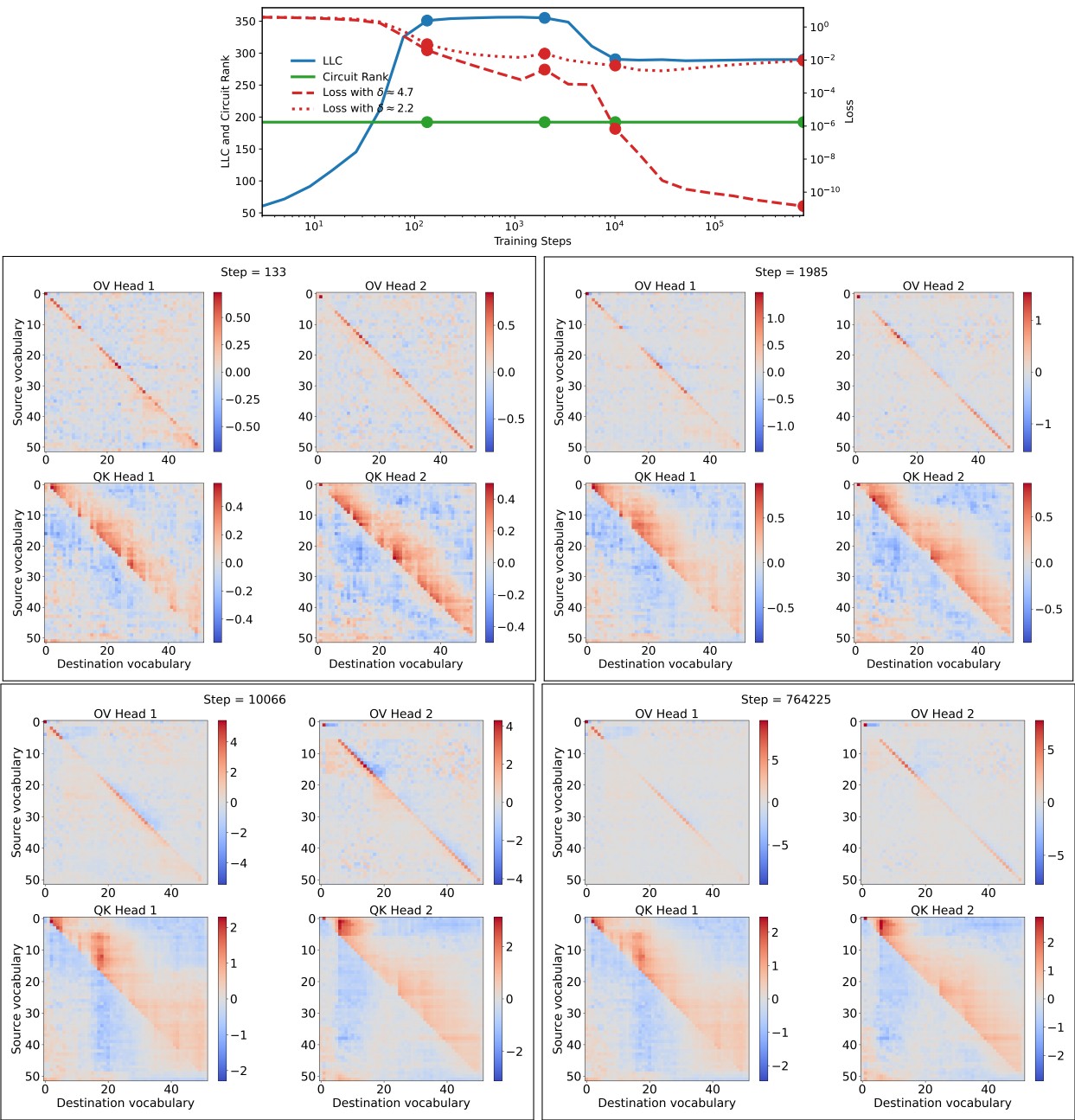

Figure 14: **Baseline 2-head model trained without WD** trained on $D_{\overline{\delta}\approx4.7}$. As the model learns how to sort (upper left), as the LLC is at its peak (upper right), after the LLC drop (lower left) and at the end of training (lower right). The loss is evaluated on $D_{\overline{\delta}\approx4.7}$ and $D^d_{\overline{\delta}\approx2.2}$.

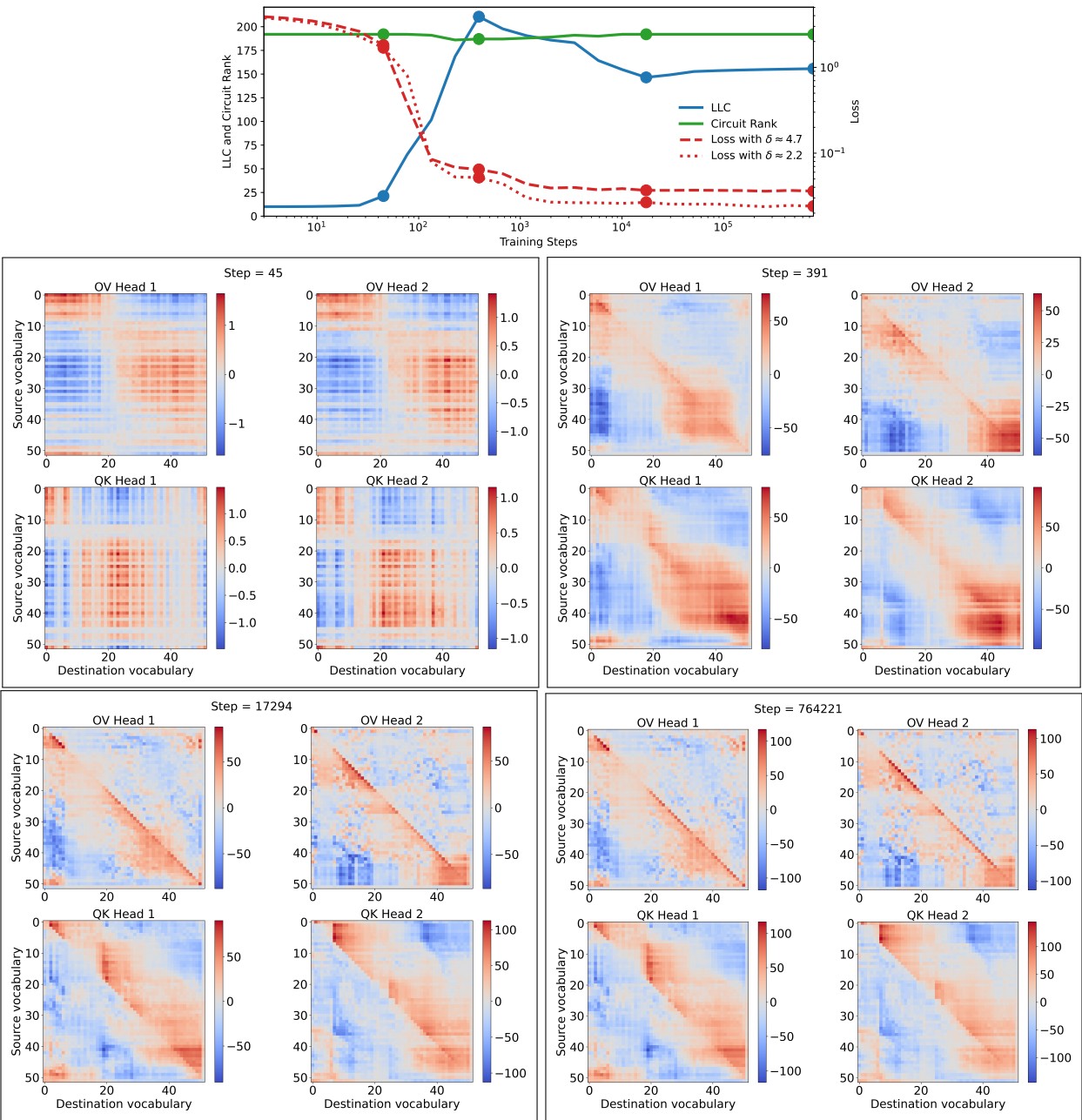

Figure 15: **Baseline 2-head model without LN and WD** trained on $D_{\bar{\delta}\approx4.7}$. As the loss starts to drop (upper left), as loss is low and LLC peaks (upper right), after LLC drop (lower left) and at the end of training (lower right). The loss is evaluated on $D_{\bar{\delta}\approx4.7}$ and $D^d_{\underline{\delta}\approx2.2}$.

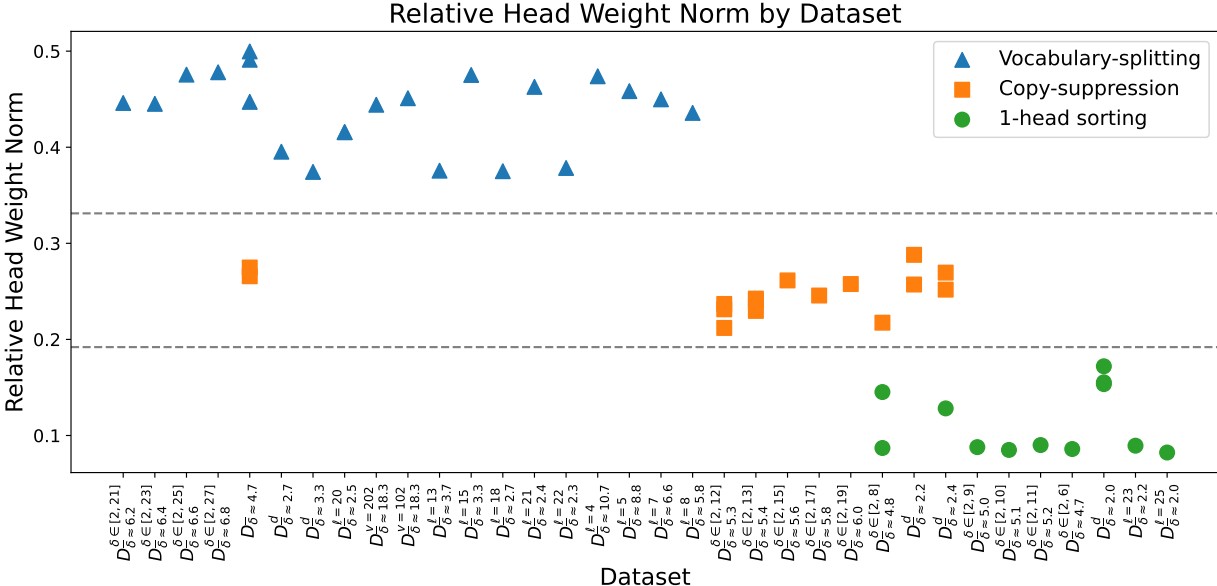

Figure 16: Distribution of relative weight norms for model heads at the end of training across different datasets. The weight norm has been computed by taking the RMS of all the weight matrices in each head (excluding the embedding and unembedding matrix). The relative weight norm is calculated by taking the weight norm of the head with the smallest weight norm and divide by the sum of the weight norms of all heads. For the 4-head model we have instead taken the weight norm of the copy-suppressing head divided by the sum of the weight norms. When a dataset has multiple markers, they correspond to different random seeds, with the exception of the left-most copy-suppressing markers which correspond to the 3-head and 4-head model, trained on $D_{\bar{\delta} \approx 4.7}$. The other markers correspond to 2-head models trained with LN and WD. The marking has been done with visual inspection of the circuits, and without consulting the relative head weight norms. We note that the different specializations separate cleanly by the horizontal lines.

Circuit Rank dipole phase, however, the model instead develops circuits that are capable of sorting, while still retaining some of the dipole-like patterns at step 391. This happens at the same time as the LLC peaks.

After this, the LLC drops, and the dipole like pattern gives way to patterns resembling the baseline 2-head model, with partial vocabulary-splitting head specialization in both QK and OV for vocabulary below around 20. We speculate that the reason why the presence of WD causes a worse performance is that it pushes the circuits into simpler low-rank dipole-like patterns instead of learning to sort. We also note that this model performs better on $D^d_{\bar{\delta} \approx 2.2}$ than on $D_{\bar{\delta} \approx 4.7}$ it was trained on.

The LLC has been calculated with inverse temperature $n\beta = 30$, step size $\epsilon = 10^{-6}$, localization term $\gamma = 56$, $n_{\text{chains}} = 4$ and $n_{\text{draws}} = n_{\text{burnin}} = 40000$.

## C    Varying the Dataset

In this subsection, we study the impact of varying aspects of the training data, such as the size of the vocabulary, the length of the list and the presence of perturbations in the data set. For a summary of all the models we trained, see Tab. 2 and Fig. 16. In the subsections we go into more details of a few models.

### C.1    Baseline 2-Head Model with Vocabulary Size Increased to 202

Increasing the vocabulary size to 202 naturally rises $\bar{\delta}$ to 18.4, and produces the training dynamics shown in Fig. 17. The model undergoes a fairly similar development as the baseline model, with the head specialization at step 2420 (lower left of Fig. 17) very similar to the baseline model trained on $D_{\bar{\delta} \approx 4.7}$. As the

Table 2: An overview of the 2-head models we train, the nature of their training dataset and the specialization they develop at the end of training: Vocabulary-splitting, Copy-Suppression or 1-head sorting. When different seeds give different specialization types, we choose the most common specialization. For specialization distribution in these cases, see Fig. 16.

| Dataset | mean $\delta$ | variance $\delta$ | List length | Vocabulary Size | Type of specialization |
|---|---|---|---|---|---|
| $D^d_{\bar{\delta}\approx2.0}$ | 2.0 | 1.6 | 10 | 52 | 1-head sorting |
| $D^d_{\bar{\delta}\approx2.2}$ | 2.2 | 2.2 | 10 | 52 | Copy-suppression |
| $D^d_{\bar{\delta}\approx2.4}$ | 2.4 | 2.7 | 10 | 52 | Copy-suppression |
| $D^d_{\bar{\delta}\approx2.7}$ | 2.7 | 3.9 | 10 | 52 | Vocabulary-splitting |
| $D^d_{\bar{\delta}\approx3.3}$ | 3.3 | 6.3 | 10 | 52 | Vocabulary-splitting |
| $D^{\delta\in[2,6]}_{\bar{\delta}\approx4.0}$ | 4.0 | 2.0 | 10 | 52 | 1-head sorting |
| $D^{\delta\in[2,8]}_{\bar{\delta}\approx4.8}$ | 4.8 | 3.9 | 10 | 52 | 1-head sorting |
| $D^{\delta\in[2,9]}_{\bar{\delta}\approx5.0}$ | 5.0 | 4.9 | 10 | 52 | 1-head sorting |
| $D^{\delta\in[2,10]}_{\bar{\delta}\approx5.0}$ | 5.0 | 5.9 | 10 | 52 | 1-head sorting |
| $D^{\delta\in[2,11]}_{\bar{\delta}\approx5.1}$ | 5.1 | 6.8 | 10 | 52 | 1-head sorting |
| $D^{\delta\in[2,12]}_{\bar{\delta}\approx5.1}$ | 5.1 | 7.6 | 10 | 52 | Copy-suppression |
| $D^{\delta\in[2,13]}_{\bar{\delta}\approx5.2}$ | 5.2 | 8.2 | 10 | 52 | Copy-suppression |
| $D^{\delta\in[2,15]}_{\bar{\delta}\approx5.2}$ | 5.2 | 9.3 | 10 | 52 | Copy-suppression |
| $D^{\delta\in[2,17]}_{\bar{\delta}\approx5.2}$ | 5.2 | 10.0 | 10 | 52 | Copy-suppression |
| $D^{\delta\in[2,19]}_{\bar{\delta}\approx5.2}$ | 5.2 | 10.5 | 10 | 52 | Copy-suppression |
| $D^{\delta\in[2,21]}_{\bar{\delta}\approx5.2}$ | 5.2 | 10.8 | 10 | 52 | Vocabulary-splitting |
| $D^{\delta\in[2,23]}_{\bar{\delta}\approx5.2}$ | 5.2 | 10.8 | 10 | 52 | Vocabulary-splitting |
| $D^{\delta\in[2,25]}_{\bar{\delta}\approx5.2}$ | 5.2 | 10.9 | 10 | 52 | Vocabulary-splitting |
| $D^{\delta\in[2,27]}_{\bar{\delta}\approx5.2}$ | 5.2 | 11.0 | 10 | 52 | Vocabulary-splitting |
| $D^{\ell=25}_{\bar{\delta}\approx2.0}$ | 2.0 | 1.9 | 25 | 52 | 1-head sorting |
| $D^{\ell=23}_{\bar{\delta}\approx2.2}$ | 2.2 | 2.3 | 23 | 52 | 1-head sorting |
| $D^{\ell=22}_{\bar{\delta}\approx2.3}$ | 2.3 | 2.6 | 22 | 52 | Vocabulary-splitting |
| $D^{\ell=21}_{\bar{\delta}\approx2.4}$ | 2.4 | 2.9 | 21 | 52 | Vocabulary-splitting |
| $D^{\ell=20}_{\bar{\delta}\approx2.5}$ | 2.5 | 3.3 | 20 | 52 | Vocabulary-splitting |
| $D^{\ell=18}_{\bar{\delta}\approx2.7}$ | 2.7 | 4.3 | 18 | 52 | Vocabulary-splitting |
| $D^{\ell=15}_{\bar{\delta}\approx3.3}$ | 3.3 | 6.5 | 15 | 52 | Vocabulary-splitting |
| $D^{\ell=13}_{\bar{\delta}\approx3.7}$ | 3.7 | 8.7 | 13 | 52 | Vocabulary-splitting |
| $D_{\bar{\delta}\approx4.7}$ | 4.7 | 14.7 | 10 | 52 | Vocabulary-splitting |
| $D^{\ell=8}_{\bar{\delta}\approx5.8}$ | 5.8 | 22.1 | 8 | 52 | Vocabulary-splitting |
| $D^{\ell=7}_{\bar{\delta}\approx6.6}$ | 6.6 | 27.8 | 7 | 52 | Vocabulary-splitting |
| $D^{\ell=5}_{\bar{\delta}\approx8.8}$ | 8.8 | 47.4 | 5 | 52 | Vocabulary-splitting |
| $D^{\ell=4}_{\bar{\delta}\approx10.7}$ | 10.7 | 65.2 | 4 | 52 | Vocabulary-splitting |
| $D^{\ell=3}_{\bar{\delta}\approx13.3}$ | 13.3 | 93.5 | 3 | 52 | Other |
| $D^{v=102}_{\bar{\delta}\approx9.4}$ | 9.4 | 63.9 | 10 | 102 | Vocabulary-splitting |
| $D^{v=202}_{\bar{\delta}\approx18.3}$ | 18.3 | 265.8 | 10 | 202 | Vocabulary-splitting |

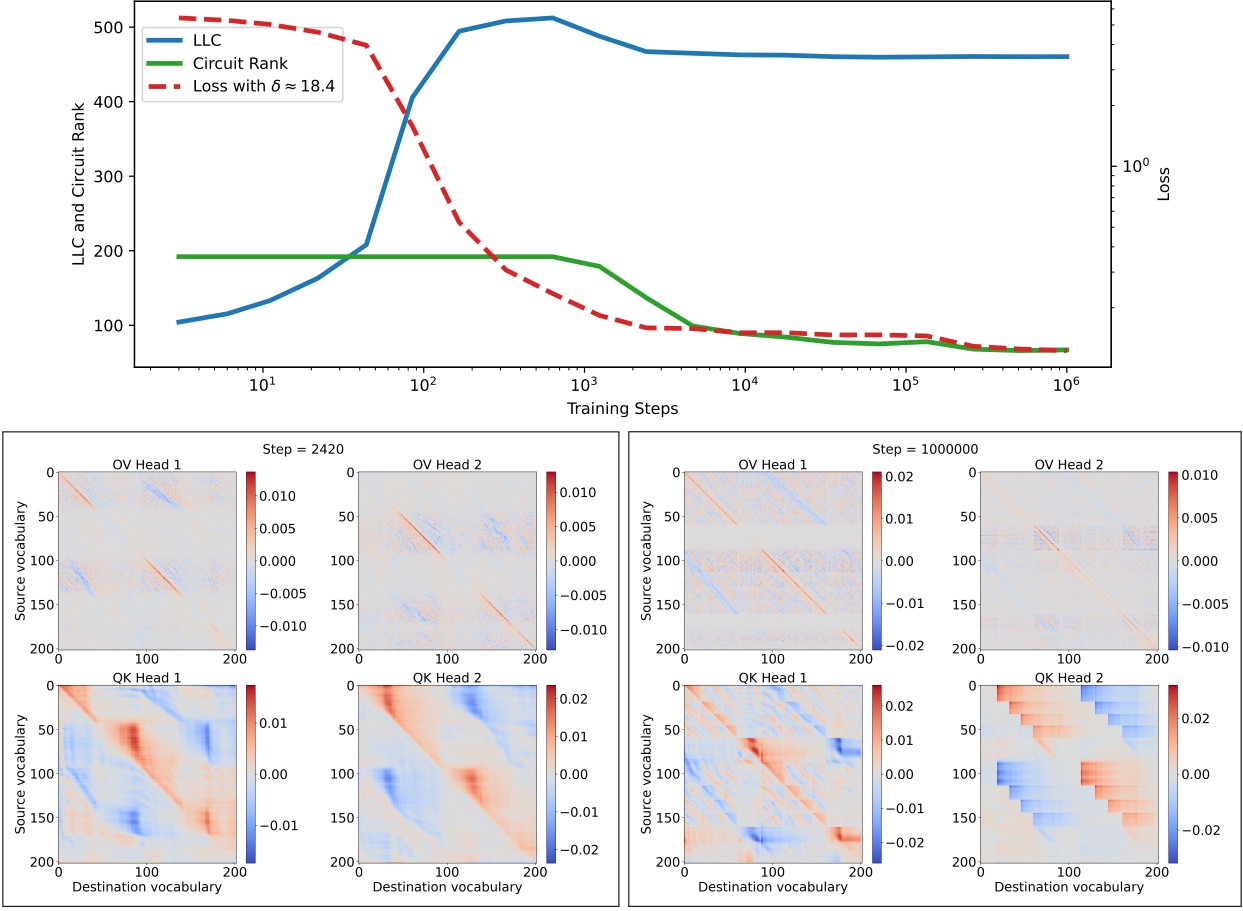

Figure 17: Baseline 2-head model with **vocabulary size increased** to 202, we find similar developmental stages as in the baseline model. **Vocabulary region size increases**. The model is trained on and the loss is evaluated on $D^{v=202}_{\delta \approx 18.3}$.

training continues, however, the model develops a square-like pattern in the QK circuit, which doesn't always correspond to a vocabulary region boundary. This last transition is accompanied by a small drop in the Circuit Rank and loss, but no drop in the LLC. During this last stage, the diagonal of the OV circuit of head 2 is positive across the entire vocabulary range, and it seems like the model has arrived at a qualitatively different solution where head 2 contributes on the entire vocabulary.

The LLC has been calculated with inverse temperature $n\beta = 512/\ln 512 \approx 82$, step size $\epsilon = 10^{-3}$, localization term $\gamma = 32$, $n_{\text{chains}} = 4$ and $n_{\text{draws}} = n_{\text{burnin}} = 2000$.

### C.2  Baseline 2-Head Model with List Length Increased to 20

Increasing the list length to 20 yields the training dynamics shown at the top of Fig. 18, with the end-of-training OV and QK circuits shown at the bottom. We note that this model stabilizes with the larger number of regions, and does not go on to have the LLC drop further and copy-suppression forming.

The LLC has been calculated with inverse temperature $n\beta = 512/\ln 512 \approx 82$, step size $\epsilon = 3 \times 10^{-6}$, localization term $\gamma = 32$, $n_{\text{chains}} = 4$ and $n_{\text{draws}} = n_{\text{burnin}} = 70000$.

### C.3  Baseline 2-Head Model with Perturbed Dataset

We perturb the data by iterating through the dataset once, and swapping neighboring elements in the sorted list with probability $40\%/(n_{i+1} - n_i)$, where $n_i$ is the value of the list element $i$. Since the probability of neighboring elements swapping is always less than 50%, we believe that the optimal strategy still should be to sort the list ignoring the perturbations but with logits more spread out. The perturbations do, however, have a severe impact on the training dynamics, as shown in Fig. 19.

We don't observe any drop in the LLC, even though The Circuit Rank does drop. The heads don't specialize into vocabulary-splitting modes, but the OV circuits rather settle into what looks like opposites of each other. It looks like head 1 does copy-suppression and head 2 does copying, whereas the QK circuits behave very differently from what we have seen in the other models.

The losses have been computed on non-perturbed data, and goes down throughout training, though it doesn't reach as low as with the baseline model.

The LLC has been calculated with inverse temperature $n\beta = 512/\ln 512 \approx 82$, step size $\epsilon = 10^{-6}$, localization term $\gamma = 32$, $n_{\text{chains}} = 4$ and $n_{\text{draws}} = n_{\text{burnin}} = 200000$.

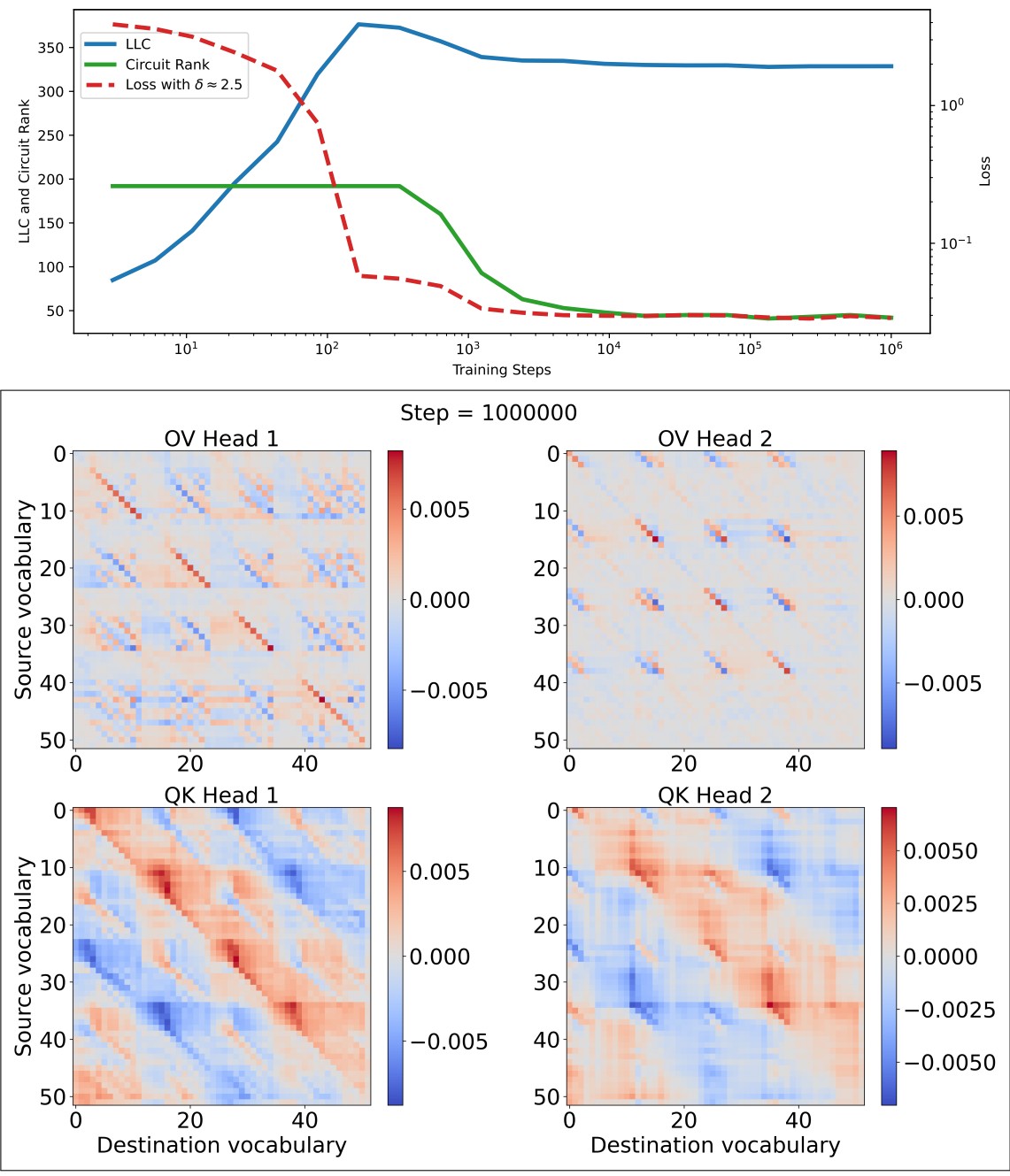

Figure 18: Baseline 2-head model with **list length increased to 20**, we find similar developmental stages as in the baseline model, but without the copy suppression. The model is trained and the loss is evaluated on $D_{\delta \approx 2.5}^{\ell=20}$.

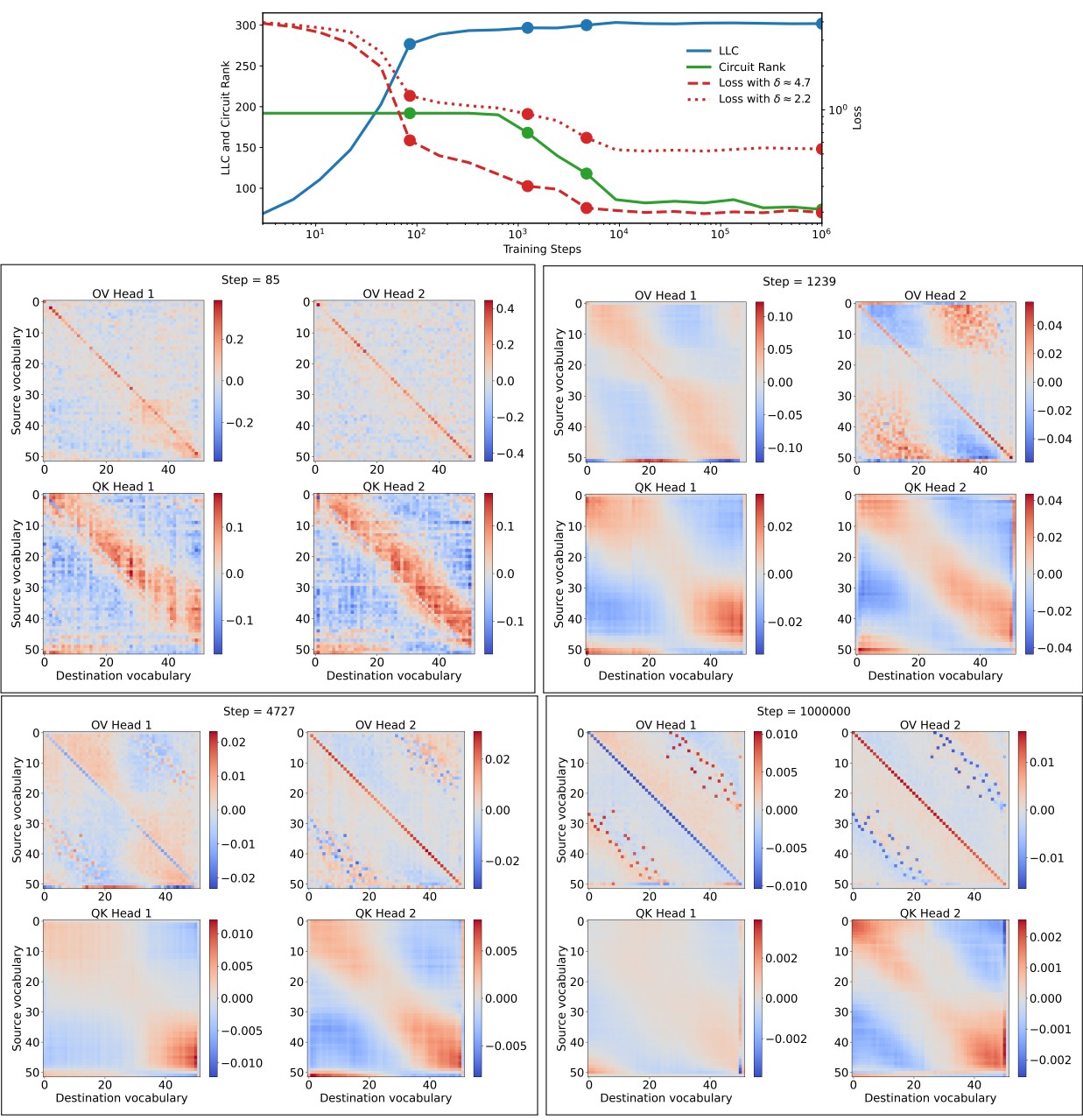

Figure 19: Baseline 2-head model trained on a **perturbed** version of $D_{\bar{\delta}\approx4.7}$. The panels show different developmental stages and it is the only 2-head model where we observe **copy-suppression.** The loss is evaluated on $D_{\bar{\delta}\approx4.7}$ and $D^d_{\bar{\delta}\approx2.2}$.

