# OpenReview forum: "Structure Development in List-Sorting Transformers"
_TMLR — Rejected by TMLR_

### Review · Reviewer_xQQs · 2024-11-28

**Summary Of Contributions:**

For a single-layer attention-only transformer, this paper looks at what kind of mechanisms attention heads develop when sorting a list from left-to-right for varying characteristics of the training data, such as list length, vocab size, list distribution, and separation between list elements in the training set. The authors find that attention heads can specialise as "vocabulary splitting" or "copy surpression", where for the former each attention head copies a different part of the vocabulary, whereas for the latter one attention head copies the entire vocabulary, and the other surpresses the entire vocabulary. Although such specialisations have been identified by prior work, the authors find that they seem driven by a metric on the training data that determines the distribution of the separation between list elements in the training set.

**Audience:**

Yes

**Broader Impact Concerns:**

N.A.

**Claims And Evidence:**

Yes

**Requested Changes:**

**Required for acceptance**:
- Improvement of clarity of the paper. I will add a list of things that I believe need clarification below.
- Motivation of the analyses and discussion of implication of findings.
- Clarification of contributions on top of prior work in the intro (this can actually already be achieved by a brief motivation in the intro if that motivation encompasses what prior work hasn't looked at yet, I would strongly consider changing the current motivation for general toy controlled studies to a motivation for the specific study done in the submission)

**Focus points for clarification**:
- What exactly is within-list separation ($\delta$)? Is it for example if you're sorting [2, 3, 1] the separation is smaller than [20, 1, 300]?
- Output-value and query-key circuits; can you briefly mention how these arise, and how they can be interpreted (i.e. the former "determines how attending to a given token affects the logits" and the latter "controls which tokens the head prefers to attend to"). For example, brief mentioning how these formulations arise and then referring the reader (explicitly) to the Transformers circuit link for detailed explanations.
- The LLC: the bullet point in the main text leaves it unclear what this metric is. Please move some details from the Appendix to the main text (can be just the interpretation, "a lower LLC indicates a more degenerate and less complex model").
- How do you separate train and validation data? Given you don't observe a difference in train versus validation loss, I am wondering how different are the data points (I presume there's no overlap)?
- What does $d$ refer to in point 3 section 2.1? I'm not sure how $D_{\bar{\delta}\approx k}$ would differ from $D^d_{\bar{\delta}\approx k}$. Is it just that the latter was created in a different way than the former, but the effective data list distribution is the same?
- "In Fig. 3 we observe that head 1 gradually covers a smaller and smaller vocabulary range, before it switches over to doing copy-suppression" -> this is mentioned in section 5.3, but while I was reading about figure 3 for the first time I didn't get that this was a gradual process you've observed.
- There is no discussion of what a copy supression head can actually be good for, why is it learned. Can you add a one-line summary of what https://arxiv.org/abs/2310.04625 find and why it would be learned/useful as well in your specific task?
- "The intuition driving this hypothesis, is that the different attention paid to two neighboring elements within an active region in the QK circuits, goes as the gradient in that region times δ. ", I could not parse this sentence. Could you explain the intuition behind the hypothesis?
- It took a long time for me to understand page 7. I have two suggestions (optional) for improvement: the first is to make a figure with a hypothetical perfect circuit that would've learned the task optimally (i.e. decreasing from left-to-right along rows from the diagonal, can be put in the appendix), and the second is to write the text explaining the figure in the caption as well or right below the figure in the text, so you can see the figure as you read it. Further, what confused me is the explanation for why low-prevalence regions are low-prevalence: "these regions only contribute if the input and output tokens correspond to different regions in the OV circuit". I still don't know what this means, but I presume the regions are lower prevalence precisely because they are further from the diagonal and it's less likely to have a next smallest number that is further from the diagonal in the data. Similarly, on the mean QK gradient metric, it would be useful to add some interpretation (i.e. if this metric is higher, it means there is a stronger gradient along rows in the right direction, meaning the model learned the task better).
- Background 3 comes a bit out of nowhere, I suspect this would be resolved by better motivating and situating in the intro
- Figure 1 requires a more informative capture that takes the reader through what's going on in this figure.
- This is a result of the issues with the intro, and again I suspect this to be resolved if the aim of the paper is made clearer in the intro, but the method section jumps straight into architectural training details and at this point the reader still essentially doesn't really know what the research question is you're trying to address and why you are doing the analysis you're doing.


**Would strengthen the submission**:
- The abstract can also use more focus on your actual findings + implications.

**Strengths And Weaknesses:**

**Strengths**.
Lots of experimentation, very clear connection found between training data properties and interpretable circuits in the model, to be more precise:
- The authors show a very clear result by training on differing in within-list element separation; if the average within-list separation is smaller, the attention heads behave differently and develop into a new specialised state that doesn't happen for the larger values of within-list separation (fig 2 versus fig 3).
- The authors hypothesise that a lower average within-list separation drives a specific specialisation shape of attention heads and show convincingly that this indeed happens (fig 6 and appendix table 2). Especially appendix table 2 is very neat.

**Weaknesses**.
The main weaknesses to this paper are all about presentation.
*Presentation*.
1. The paper is incredibly hard to follow, uses a lot of specialised terminology that is not explained, and misses details that help understand the paper (which can be taken from prior work or are just simply missing, such as how a training set versus validation set is chosen for train versus val loss)
2. A lot of experiments that drive the contribution are in the appendix. This is not necessarily a problem in and of itself, but the explanations in the main text about these experiments are very brief and did not help me understand them without reading the appendix. For example, I did not fully grasp how well the mean $\delta$ parameter determines the mode of attention head specalisation the model settles into until I read the appendix.
3. The paper can really use some work on presentation of the findings, why we should care about this problem, and what implications do the findings have. There is a 4-paragraph discussion in the introduction on why we need to do controlled studies on toy tasks for interpretability research more generally, and only 1 paragraph on the actual method and results combined. There is no discussion of the implications of the findings, or a situation w.r.t. prior work / motivation for this specific investigation in the introduction.

*Contribution*.
1. The contribution can be made clearer in the paper, as for me currently it's not entirely clear. I think the contribution is that we learned above and beyond prior work what drives specialisation of attention heads in terms of architecture/training setup (weight decay) and most of all aspects of the training data (within-list separation distribution), but there is no discussion of why we want to know this, how these findings can help us moving forward, and how these could translate to more realistic setups.

*Soundness*.
The soundness seems good, but at times this is hard to know for sure due to missing details, for example but see more in requested changes below:
1. how training is split from validation data.
2. For Figure 6 left, is the coefficient found with LR significant? From the plot it seems like there is really not a strong downward slope with $\bar{\delta}$

---

> ### Author Response · Authors · 2024-12-22
> **Response addressing required changes and suggestions.**
>
> Dear Reviewer,
>
> Thank you very much for your detailed review and helpful inputs, and apologies for our late response. We have attempted to address your feedback and will upload a new version of the paper with changes marked in red. As the length of this comment is constrained, we will not go through every risen point separately, but we have tried to address all of them in the new version of the paper.
>
> We have reworked:
> * the abstract, to highlight our main findings better
> * the introduction, to improve the motivation of the analysis for our specific model.
> * the discussion, to clarify how our contributions relate to prior work and implications. In particular, for copy-suppression we add some minor experiments that clarify the role of this specialization.
> * the conclusions, to improve our statement of contributions and implications of our findings.
>
> We have tried to address all the points of focus for clarification.
>
> Regarding the raised weaknesses of the presentation:
> * We agree that we use a lot of specialized terminology. We have tried to mend this where it was pointed out to us, but we are aware that the specialized terminology still makes our paper somewhat inaccessible.
> * We chose to keep a lot of the experiments in the appendix to constrain the length of the main body of the paper. We have attempted to mend this by expanding our discussion of the way $\delta$ drives the size of the regions.
>
> Regarding the soundness:
> * We have explicitly checked that the training and validation data has no overlap, and have clarified this in the paper.
> * Regarding figure 6, the coefficients found with linear regression are meant to emphasize the flatness of the slope and complement the visual interpretation of the figure.

---

> > ### Comment · Reviewer_6ka6 · 2024-12-30
> >
> > I thank the authors for making substantial revisions to the paper to try to present its claims more clearly. My assessment of the paper, however, is still that the paper needs to do much more to explain its claims and the significance of those claims to a TMLR audience. If the other reviewers and action editor believe that it's sufficiently accessible, I'm happy to defer to their judgement. But my recommendation would be to enlist the help of someone outside of the MI community to identify where the paper needs to be expanded or rewritten.
> >
> > In the list of contributions,
> >
> > - You write that "vocabulary-splitting...is a less complex solution that is not driven by the presence of weight-decay in training." Less complex than what? Why would the reader think that it was driven by weight-decay? Why is it important that it is not?
> >
> > - You write that "copy-suppressing heads...counteract the copying circuits" but don't explain why they form, or why they are interesting. You write that they "adjust or calibrate the copying head" but I don't understand what these terms mean. This paragraph is very dependent on McDougall 2023. Why is it important that it increases rather than reduces confidence?
> >
> > - You identify that the distribution over the difference of adjacent list elements affects head specialization and that you "gain insight on the important role that the latent features have," but what are those insights?

---

> > > ### Author Response · Authors · 2025-01-02
> > >
> > > We thank the reviewer for their time and the detailed review.
> > >
> > > We have expanded and rewritten the introduction, the abstract and the conclusion to try to make it more accessible to an audience without a background in mechanistic interpretability.
> > >
> > > * We have clarified that vocabulary-splitting is less complex than the stage coming before it where the two attention heads cover overlapping vocabulary.
> > > * We have clarified in the introduction that weight decay is believed to be responsible for driving the neural network to simplify. We find that the network simplifies even without weight decay, strengthening theoretical predictions (for instance from SLT) that networks should naturally prefer simpler solutions.
> > > * We have tried to expand upon and clarify the significance of our observation of copy-suppression. We think that our finding that  copy-suppression increases confidence is interesting because it points towards it being a broader phenomenon of adjustments. By adjustments we mean that the output of the copy-suppressing head is sub-leading to the output of the copying head, so that when added together the output of the copy-suppressing head serves to make small changes (adjustments) to the output of the copying head. This results in a (comparatively) small change in the logit distribution if the head is ablated.
> > > * We agree that the last point of the contributions could have been formulated better, and have now rewritten it to emphasize that our study paves the way for future studies considering the impact features in the data has on structures the model learns.

---

> > ### Comment · Reviewer_xQQs · 2025-01-05
> > **Thanks for the revision**
> >
> > I went over all my points that needed to be clarified in the revision, and found them all sufficiently addressed. Thank you for this. I especially like the updated discussion on how your findings give some evidence for the universality hypothesis, I had not thought of that on the first read.
> >
> > All in all, I would still suggest the authors go over aspects of the paper to update some things, but these are unrelated to the content and contributions of the work. E.g. sometimes citations aren't properly done (without brackets when brackets are required, with double brackets, etc.).
> >
> > Taken together, although this paper studies a very simple toy task in a model that is not used in practice, I think it gives insights into what kind of structures can be learned when, how they depend on training data properties, and provide some evidence for the universality hypothesis.

---

### Review · Reviewer_6ka6 · 2024-11-30

**Summary Of Contributions:**

This paper analyzes the weights of transformers trained on variations of a list-sorting task. It identifies three or four phases in training: initial learning, head-overlapping, vocabulary-splitting, and copy-suppression.

**Audience:**

Yes

**Claims And Evidence:**

No

**Requested Changes:**

Please explain much more explicitly and clearly what vocabulary-splitting and copy-suppression are, how you see them in the QK and OV heatmaps shown in the figures, and why these behaviors are of interest.

**Strengths And Weaknesses:**

I had difficulty understanding this paper, and it's possible I'm not the right reviewer for it. The first TMLR evaluation criterion is whether a paper supports the claims that it makes, but to me the claims of the paper are not clearly stated. Vocabulary-splitting and copy-suppression are not defined, it's not explained what patterns in the QK and OV heatmaps are indicative of these behaviors, and it's not clear what insights are gained by identifying these behaviors.

---

> ### Author Response · Authors · 2024-12-23
> **Response to address remarks**
>
> Dear Reviewer,
>
> Thank you for your review and apologies for the late reply. We are aware that the paper is somewhat inaccessible and has a lot of specific terminology. We have tried to improve this in a new version of the paper, with changes marked in red.
>
> Regarding the requested changes:
> * We have improved our definitions for the OV and QK circuits, which form the basis for defining vocabulary-splitting and copy-suppression. We hope this clarifies the definitions for these head specializations.
> * We have added additional context and discussion on why these head specializations are of interest, in Sec. 4.2 and 4.3. In particular, copy-suppression is interesting because it might be related to a state observed in GPT-2, a much more complex model than our own. We present some arguments in favor of this and some key differences. If true, this supports the universality hypothesis. We compare the role of copy-suppression in our model with the role it plays in GPT-2. On the other hand, vocabulary-splitting is interesting to us for more technical reasons: it's an example of a head specialization that balances simplicity (low LLC and Circuit Rank) and generality for list sorting (since it is trained on a more diverse dataset, meaning higher $\delta$ variance).
>
> Additionally, we have rewritten major parts of the paper's introduction and conclusions to improve the motivation of analysis and our contributions on top of prior work.

---

### Review · Reviewer_riyZ · 2024-12-09

**Summary Of Contributions:**

The paper analyzes the training dynamics of a single-layer attention-only transformer trained to sort lists of numbers. The paper investigates the impact of the distribution of separations (δ) between list elements on head specialization. They hypothesize and provide evidence that smaller δ necessitates larger differences in the attention heads, influencing region size and the number of regions in vocabulary splitting. They also connect the variance of δ to the emergence of different specialization types. They identify distinct developmental stages characterized by changes in attention weights and loss. The study reveals two specialization modes for the attention heads: vocabulary-splitting, where heads partition the vocabulary among themselves, and copy-suppression, where one head counteracts the copying behavior of another. Copy-suppression is presented as a novel minimal example of this phenomenon, previously observed in more complex models like GPT-2.

**Audience:**

Yes

**Broader Impact Concerns:**

No ethical implications. Purely theoretical study.

**Claims And Evidence:**

Yes

**Requested Changes:**

The paper lacks mathematical rigor in many places. For example:
- "distribution of the separation between list elements in the training dataset" - how is it precisely defined?
- "we shall denote a dataset D with mean δ = x as D_{δ=x}, where we denote the mean δ as $\bar δ$" - I'm not sure what this means either.
- "For reference, the procedure described above produces..." - which procedure?
- There is no formal definition of LLC. The formula in Appendix A doesn't actually define anything, as $t$ depends on $\beta$ and $\gamma$ in some unknown way. Also, what is $n$ in $l_n$?

What happens if you fine-tune one model with data from another distribution? Will the model find the other solution? What are the stages that the model go through in this case?

Overall, how general is the final trained model? For example, does the final trained model generalize to a longer list length? Shorter?

What happens when you increase the vocabulary size?

How susceptible the developmental stages to change in the optimization hyperparameters?

**Strengths And Weaknesses:**

Strengths:
- Connection between the data distribution and head specialization in a list-sorting transformer. Novel insights into its developmental stages, including the discovery of copy-suppression in a simplified model.
- The paper situates itself well within the existing literature on mechanistic and developmental interpretability.

Weaknesses:
- The paper lacks precise mathematical definitions and formulas for key concepts.
- The study focuses on a very simplified toy model. While this offers a controlled environment for analysis, the generalizability of the findings to more complex transformers remains an open question. The authors acknowledge this limitation.
- The paper focuses primarily on the OV circuit for explaining developmental stages, while QK circuits are mostly in the the appendix.
- Some figures (like Fig. 15) could benefit from better labeling and clearer explanations in the caption. It would be better to combine Fig.2 and Fig.3 side-by-side for easier visualization of the results.

---

> ### Author Response · Authors · 2024-12-23
> **Response addressing required changes and suggestions.**
>
> Dear Reviewer,
>
> Thank you for your helpful review and feedback, and apologies for the late response. We have addressed your requested changes in a new version of the paper, with changes marked in red. In addition, we found your suggestion for further fine tuning of the model interesting and included this as well.
>
> Regarding the requested changes:
> * we have expanded upon the definition of $\delta$ and clarified how the datasets are generated
> * we have expanded the appendix on the LLC to include the formal definitions of all relevant quantities that appear.
> * we conduct "fine-tuning" experiments and train two of our main setups further to see whether they will transition out of the head specialization they are in. We find that the baseline 2-head model that develops vocabulary-splitting transitions to copy-suppression when further trained on $D^d_{\overline{\delta}}\approx 2.2$. On the other hand, the model from Fig. 3 in the paper that develops copy-suppression, doesn't transition to vocab-splitting or any other state.
> * regarding the generality of the final state of the model, vocabulary-splitting constitutes the more general mode, since it comes about when training on a more diverse dataset (meaning larger variance in the $\delta$ distribution). It performs better out-of-distribution than copy-suppression, for example, which is trained on less diverse datasets.
> * experiments related to increasing vocabulary size corresponds to point 2. in the list in Sec. 2.1, in which we show how we vary the training dataset. Increasing vocabulary for fixed list-length, increases the mean $\delta$, which in turn drives the region size up.
> * We have experimented with removing weight decay, layer norm and both at the same time. We find that vocabulary-splitting is robust with respect to these, although it becomes significantly more noisy. These experiments are in the Appendices B.5-B.7. We improved the discussion section 4.2 to highlight this point better.
>
> Regarding the weaknesses
> * we have tried to address the mathematical rigorousness of the LLC definition in the Appendix
> * We chose to move the QK circuits to the Appendix and focus on the OV circuits in the main body to reduce the content overload and make the paper easier to follow.
> * We have expanded the introduction of the notation of the datasets, hoping that this will clarify Fig. 15. We have chosen not to place Figs. 2 and 3 side by side as this would require to make the figures smaller.

---

> > ### Comment · Reviewer_riyZ · 2025-01-03
> >
> > Thanks for the answer. I still find the paper not easy to understand. For example, the definition of $\delta$ remains unclear and lacks mathematical rigor. The provided explanation, "Regarding the latter, we find that most of the impact of varying the training data can be boiled down to changes in the distribution of the separation between list elements in the training dataset, denoted by δ. For example, [8, 3, 5, SEP, 3, 5, 8] has the δ values [2, 3]", is informal and ambiguous. It fails to precisely define what constitutes "separation" and how the values of $\delta$ are derived.

---

> > > ### Author Response · Authors · 2025-01-03
> > >
> > > Thanks for the comment, we have clarified the definition of $\delta$ by providing explicit mathematical expression for how we calculate $\delta$ and $\overline{\delta}$. These changes can be found on page 4 of the paper. If there are other specific improvements or clarifications we could make in the paper we would be happy to address them.

---

### Decision · Action_Editor_hJta · 2025-01-24

**Recommendation:** Reject

**Comment:**

The reviewers agree that the work is good and can be published at TMLR. One of the reviewers disagrees but also admits to being a bit outside of the field, and hence, on this point, I think the judgement of the two "expert" reviewers can prevail.

However, at least two of the reviewers, even after the discussion and the significant improvements to the manuscript, find the latest manuscript requires substantial revisions to make it accessible to the TMLR audience. One issue is that they only make a few broad suggestions (e.g., one reviewer mentions "improve mathematical rigour throughout the paper" and another that the authors should further explain the "significance of the claims").

As a result, it's a bit difficult to judge how much work is required, but I would like reviewers to judge an updated version of the paper before it is accepted. The TMLR mechanism for doing this is to "reject the paper" with the suggestion to resubmit a revision later. If the authors decide to do this, the hope is that the same reviewers (at least the two experts) could be enlisted, which might lead to an expedited process.

**Audience:**

The reviewers all agree that the work would interest the TMLR audience.

**Claims And Evidence:**

According to the reviewers, the latest manuscript revision has improved, and two out of three reviewers now believe the claims are well supported by evidence. The third reviewer's arguments indicate that the paper remains tough to understand and follow. Still, this might relate more to the second criterion (whether the current manuscript interests the audience).

**Resubmission Of Major Revision:**

The authors may consider submitting a major revision at a later time.